# RNA $N^6$-methyladenosine modulates endothelial atherogenic responses to disturbed flow in mice

Bochuan Li[1], Ting Zhang[2,3,4], Mengxia Liu[2,3,4], Zhen Cui[1], Yanhong Zhang[1], Mingming Liu[1], Yanan Liu[1], Yongqiao Sun[2,3], Mengqi Li[1], Yikui Tian[1], Ying Yang[2,3,4]*, Hongfeng Jiang[5]*, Degang Liang[1]*

[1]Tianjin Key Laboratory of Metabolic Diseases, Key Laboratory of Immune Microenvironment and Disease (Ministry of Education), Collaborative Innovation Center of Tianjin for Medical Epigenetics and Department of Physiology and Pathophysiology, Department of Cardiovascular Surgery, Tianjin Medical University General Hospital, Tianjin Medical University, Tianjin, China; [2]CAS Key Laboratory of Genomic and Precision Medicine, Collaborative Innovation Center of Genetics and Development, College of Future Technology, Beijing Institute of Genomics, Chinese Academy of Sciences, Beijing, China; [3]China National Center for Bioinformation, Beijing, China; [4]University of Chinese Academy of Sciences, Beijing, China; [5]Key Laboratory of Remodeling-Related Cardiovascular Diseases (Ministry of Education), Beijing Collaborative Innovation Center for Cardiovascular Disorders, Beijing Institute of Heart Lung and Blood Vessel Diseases, Beijing Anzhen Hospital, Capital Medical University, Beijing, China

**\*For correspondence:**
yingyang@big.ac.cn (YY);
jhf@pku.edu.cn (HJ);
15922230066@163.com (DL)

**Competing interest:** The authors declare that no competing interests exist.

**Abstract** Atherosclerosis preferentially occurs in atheroprone vasculature where human umbilical vein endothelial cells are exposed to disturbed flow. Disturbed flow is associated with vascular inflammation and focal distribution. Recent studies have revealed the involvement of epigenetic regulation in atherosclerosis progression. $N^6$-methyladenosine (m$^6$A) is the most prevalent internal modification of eukaryotic mRNA, but its function in endothelial atherogenic progression remains unclear. Here, we show that m$^6$A mediates the epidermal growth factor receptor (EGFR) signaling pathway during EC activation to regulate the atherosclerotic process. Oscillatory stress (OS) reduced the expression of methyltransferase like 3 (METTL3), the primary m$^6$A methyltransferase. Through m$^6$A sequencing and functional studies, we determined that m$^6$A mediates the mRNA decay of the vascular pathophysiology gene *EGFR* which leads to EC dysfunction. m$^6$A modification of the *EGFR* 3' untranslated regions (3'UTR) accelerated its mRNA degradation. Double mutation of the *EGFR* 3'UTR abolished METTL3-induced luciferase activity. Adenovirus-mediated METTL3 overexpression significantly reduced EGFR activation and endothelial dysfunction in the presence of OS. Furthermore, thrombospondin-1 (TSP-1), an EGFR ligand, was specifically expressed in atheroprone regions without being affected by METTL3. Inhibition of the TSP-1/EGFR axis by using shRNA and AG1478 significantly ameliorated atherogenesis. Overall, our study revealed that METTL3 alleviates endothelial atherogenic progression through m$^6$A-dependent stabilization of *EGFR* mRNA, highlighting the important role of RNA transcriptomics in atherosclerosis regulation.

## Editor's evaluation

Methylation of adenine residues in mRNA has been shown to be a regulator of many factors in heath and disease. In these studies, the authors present data that this modification of the mRNA for EGFR

(epidermal growth factor receptor) through down regulation of the methylating enzyme METTL3 by shear stress is a contributor to vascular pathology in a model with some features of accelerated atherosclerosis.

## Introduction

$N^6$-methyladenosine (m$^6$A) is the most prevalent post-transcriptional modification of eukaryotic mRNAs (*Roundtree et al., 2017*). This modification is reversible and is catalyzed by a multicomponent methyltransferase complex consisting of various methyltransferases including methyltransferase like 3 (METTL3), METTL14, Wilms tumor 1-associated protein (WTAP), and KIAA1429 (Virilizer), and is erased by demethylases such as fat mass and obesity-associated protein or α-ketoglutarate-dependent dioxygenase alk B homolog 5 (*Shi et al., 2019*; *Huang et al., 2021*). As previously reported, m$^6$A on mRNAs plays an important role in regulating cellular processes, including RNA stability, translation efficiency, RNA secondary structure, subcellular localization, alternative polyadenylation, and splicing (*Liu et al., 2015*; *Wang et al., 2015*; *Zaccara et al., 2019*). In the methyltransferase complex, METTL14 functions as the target recognition subunit by binding to RNA and then recruits METTL3 to catalyze m$^6$A formation. Wilms tumor 1-associated protein (WTAP) is a regulatory subunit required for the accumulation of METTL3 and METTL14 into nuclear speckles (*Ping et al., 2014*). KIAA1429 guides region-selective m$^6$A methylation (*Yue et al., 2018*). As the core methyltransferase subunit, METTL3 has been demonstrated to modulate key physiological processes, including spermatogenesis (*Xu et al., 2017*), cell reprogramming (*Chen et al., 2015*), and embryonic stem cell chromatin modification (*Liu et al., 2021*; *Xu et al., 2021*). However, its function in human cardiovascular disease (CVD) remains elusive.

Atherosclerosis, resulted from endothelial dysregulation in the arterial wall (*Geovanini and Libby, 2018*), is the leading cause of CVD resulting in high rate of mortality in the population. Atherosclerosis preferentially develops at branches and curvatures in the arterial tree where flow is disturbed (*Davies, 2009*). Disturbed flow pattern increases inflammatory response in ECs, including the expression of intercellular adhesion molecule 1 and vascular adhesion molecule 1 (VCAM-1) (*Humphrey et al., 2014*). Recent studies have reported that oscillatory stress (OS) could modulate atherosclerosis development by inducing the expression of DNA methyltransferases (*Dunn et al., 2014*) and histone modifications (*Hastings et al., 2007*), indicating the involvement of epigenetic mechanisms in the regulation of atherogenesis. In this study, we focused on exploring the function of RNA m$^6$A modification under OS in ECs, which may provide a better understanding of atherogenesis.

Numerous studies have shown that thrombospondin-1 (TSP-1, encoded by *THBS1*) is a shear-sensitive protein important for the regulation of vascular remodeling (*Ni et al., 2010*). Notably, TSP-1 can induce dysregulated blood flow, impaired vessel dilation, and increased vascular tone *Csányi et al., 2012* followed by arterial stiffening (*Kim et al., 2017*). In addition, *Thbs 1* deficiency prevented lesion formation in *Apoe$^{-/-}$* mice (*Ganguly et al., 2017*). As a receptor of TSP-1, epidermal growth factor receptor (EGFR) has been reported to be involved in vascular pathophysiology and pathogenesis of atherosclerosis in macrophages (*Wang et al., 2017*). Moreover, EGFR-selective tyrphostin, AG1478, can reverse the phosphorylation of EGFR tyrosine-1068 induced by TSP-1 activation (*Liu et al., 2009*), indicating the complicated regulation between TSP-1 and EGFR with respect to vascular function.

In this study, we generated tamoxifen-inducible endothelial-specific *Mettl3*-deficient (EC-*Mettl3$^{KO}$*) mice to investigate the regulation of *Mettl3*-mediated m$^6$A on atherogenesis. Our data revealed that downregulation of METTL3 and hypomethylation mediate OS-induced endothelial dysfunction and atherogenesis both in vivo and in vitro, indicating that the *THBS1/EGFR* axis is a key regulatory target of METTL3-dependent EC activation. These results illuminate a critical mechanism of m$^6$A modification in regulating atherosclerosis.

## Results

### METTL3 is decreased in atheroprone regions

To explore the functions of m$^6$A modification under OS, we first detected m$^6$A level changes in response to OS (0.5 ± 4 dyn/cm$^2$, 1 Hz) for 6 hr in human umbilical vein ECs (HUVECs) through UHPLC-MRM-MS

(ultra-high-performance liquid chromatography-triple quadrupole mass spectrometry coupled with multiple-reaction monitoring) analysis. The results showed that OS significantly decreased m⁶A modification in HUVECs (***Figure 1A***). To further investigate the effects of OS on the pattern of m⁶A modulators, we examined methyltransferase components including METTL3, METTL14, METTL16, WTAP, and Virilizer. Western blot analysis detected a significantly decrease of METTL3 and Virillizer after OS stimulation, without affecting other m⁶A modulators (***Figure 1B–E***). Furthermore, to analyze the m⁶A modulators in atheroprone (aortic arch [AA]) and atheroprotective (thoracic aorta [TA]) regions, we performed western blot analysis and found that METTL3 and Virilizer were significantly decreased in AA tissue lysates than in TA region, whereas METTL14 and WTAP were not changed (***Figure 1—figure supplement 1A and B***). Consistently, in outer curvature of AA and TA, where blood flow is laminar, METTL3 was highly expressed in both regions but only displayed specific nuclear localization in outer curvature of AA. Conversely, in the inner curvature and bifurcation of AA, where blood flow is disturbed, METTL3 was dramatically decreased (***Figure 1F and G***; ***Figure 1—figure supplement 1C***). Next, to further test the expression of METTL3, we used *Apoe⁻ᐟ⁻* mice with partial ligation and performed carotid Doppler ultrasonography to verify disturbed flow induction in the partially ligated left common carotid artery (LCA) at 7ᵗʰ and 14ᵗʰ days after the procedure compared with stable flow in those at day 0 (***Figure 1—figure supplement 1D***). Consistent with the AA, as compared with right common carotid artery (RCA), METTL3 was weakly expressed in LCA after 1 and 2 weeks (***Figure 1—figure supplement 1E and F***). Collectively, METTL3 is decreased in atheroprone regions.

## Endothelial activation arises in EC-specific METTL3-deficient mice

Next, to clarify whether METTL3 deficiency induces endothelial dysfunction, we crossed tamoxifen-inducible endothelial-specific *Mettl3*-deficient (EC-*Mettl3^KO*) mice (***Figure 2—figure supplement 1A*** and B). METTL3 was used as a marker of knockout efficiency. The protein levels of METTL3 were decreased in AA intima tissue lysates compared to TA in *Mettl3^flox/flox* mice. The level of the EC activation marker VCAM-1 was significantly increased in AA intima tissue lysates compared to TA intima lysates in both *Mettl3^flox/flox* mice and EC-*Mettl3^KO* mice (***Figure 2A–C***). Next, we used EC-*Mettl3^KO* mice and *Mettl3^flox/flox* mice with partial ligation to further test the expression of METTL3. En face immunofluorescence staining of LCA revealed reduced protein levels of METTL3 and enhanced levels of VCAM-1 in EC-*Mettl3^KO* mice compared to RCA 2 weeks after ligation (***Figure 2D and E***). Overexpression of METTL3 in ECs by adeno-associated virus (AAV9-METTL3 OE) inhibited VCAM-1 expression induced by partial ligation in LCA endothelium (***Figure 2F and G***; ***Figure 2—figure supplement 1C and D***). These results indicate that METTL3 depletion is associated with EC activation in response to OS.

## OS-abolished m⁶A prevents *EGFR* mRNA degradation

As a core subunit of the m⁶A methyltransferase complex, the downregulation of METTL3 expression in response to OS suggests the potential regulation of m⁶A modification. First, we conducted m⁶A-specific methylated RNA immunoprecipitation combined with high-throughput sequencing (MeRIP-seq) to compare the landscape of m⁶A in static treatment (ST) and OS. We identified 10,515 and 10,580 m⁶A peaks in ST and OS, respectively, all of which were enriched in coding regions, 3′ untranslated regions (3′UTRs), and near stop codons (***Figure 3A***; ***Figure 3—figure supplement 1A***). Interestingly, we found that the m⁶A motifs identified in OS and ST were significantly enriched in GGACU (***Figure 3B***). Furthermore, we identified thousands of dysregulated m⁶A peaks induced by OS (***Figure 3—figure supplement 1B***; ***Supplementary file 1***), and genes with downregulated m⁶A peaks were significantly enriched in transcription regulation- and cell-cell and membrane adhesion-related pathways (***Figure 3—figure supplement 1C***, upper panel). Overall, we found that METTL3 may mediate the dynamic change in m⁶A landscapes between ST and OS.

As reported, m⁶A modification plays a very important role in regulating RNA abundance (***Wang et al., 2014***). To investigate the molecular mechanisms of m⁶A function in endothelial activation, RNA-seq was performed using ECs treated with ST and OS. We identified 547 consistently upregulated genes (***Figure 3—figure supplement 1D-F***) upon OS. Consistent with the MeRIP-seq data, these upregulated genes were also significantly enriched in transcription regulation- and cell adhesion-associated pathways (***Figure 3—figure supplement 1G***, right panel). Therefore, we hypothesized that METTL3 regulates the degradation of these genes in response to OS. Furthermore, we analyzed

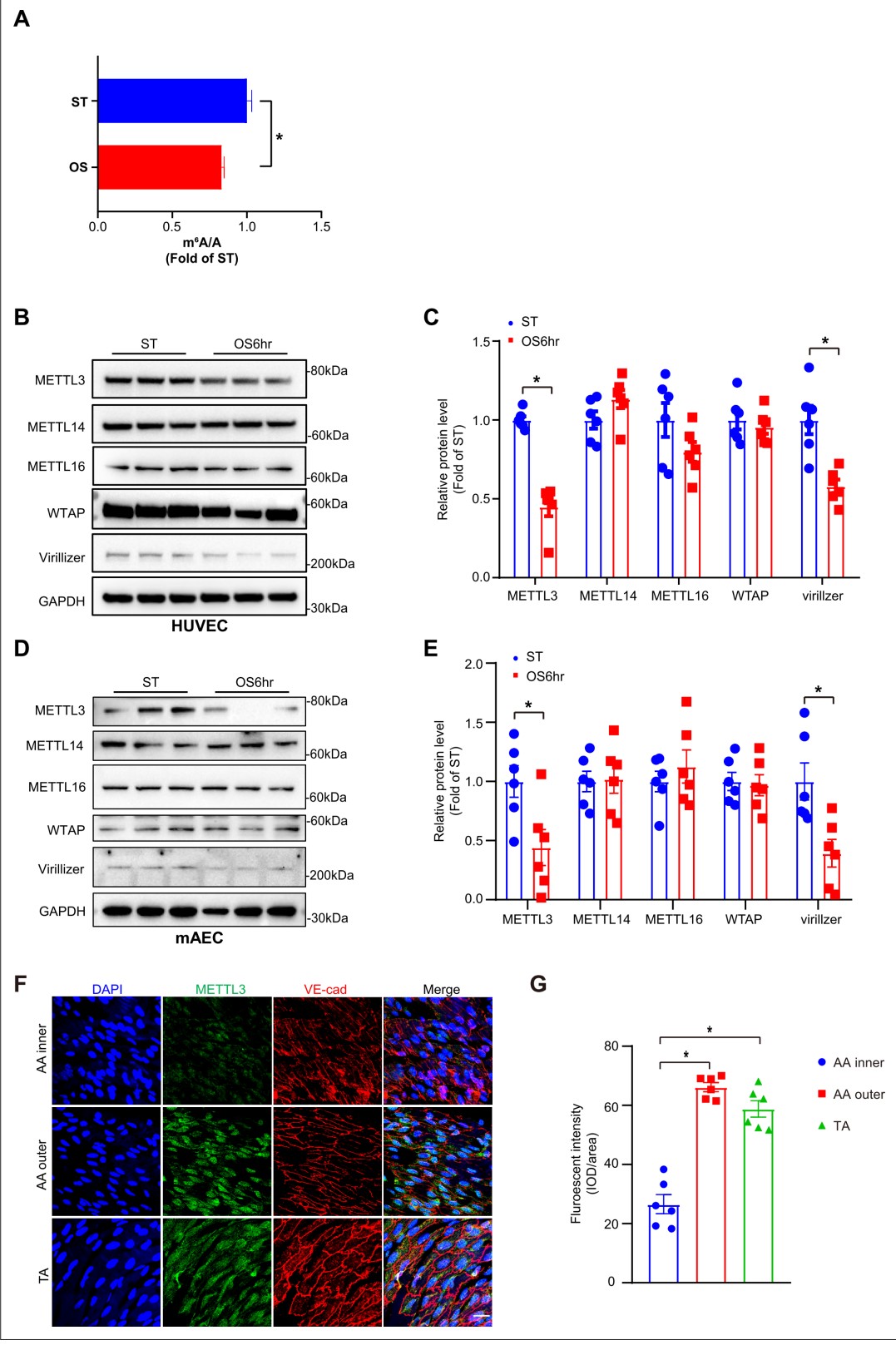

**Figure 1.** Methyltransferase like 3 (METTL3)-dependent $N^6$-methyladenosine (m$^6$A) methylation is decreased in atheroprone regions. Human umbilical vein endothelial cells (HUVECs) and mouse aortic endothelial cells (mAECs) were exposed to OS ($0.5 \pm 4$ dyn/cm$^2$) for 6 hr. Cells with static treatment (ST) were a control. (**A**) Ultra-high-performance liquid chromatography-triple quadrupole mass spectrometry coupled with multiple-reaction

*Figure 1 continued on next page*

*Figure 1 continued*

monitoring analysis of m⁶A levels in mRNAs extracted from HUVECs exposed to ST and OS. Data are shown as the mean ± SEM, *p<0.05, NS, not significant (Student's *t* test). n = 3. (**B–E**) Western blot analysis of METTL3, METTL14, METTL16, Wilms tumor 1-associated protein, and Virillizer expression in HUVECs (**B–C**) and mAECs (**D–E**) response to ST and OS. Data are mean ± SEM, *p<0.05 (Student's *t* test). n = 6. (**F**) Aortas from 6- to 8-week-old *Apoe*⁻/⁻ mice underwent immunofluorescence staining for indicated proteins. AA inner, inner curvature of aortic arch; AA outer, outer curvature of aortic arch; TA, thoracic aorta. Scale bar, 20 µm. (**G**) Quantification of protein expression in (**F**). Data are mean ± SEM, *p<0.05 (one-way ANOVA with Bonferroni multiple comparison post-test). n = 6.

The online version of this article includes the following source data and figure supplement(s) for figure 1:

**Source data 1.** METTL3-dependent m⁶A methylation is decreased in atheroprone regions.

**Figure supplement 1.** Methyltransferase like 3 (METTL3) is decreased in atheroprone regions.

**Figure supplement 1—source data 1.** METTL3 is decreased in atheroprone regions.

---

the upstream regulators of methylation-downregulated but expression-upregulated genes using IPA software. According to the IPA analysis, we identified *EGFR* as a potential key regulator participating in OS-enhanced gene ontology (GO) pathways (*Figure 3—figure supplement 1D*; *Supplementary file 2*), which was further confirmed by the reduced ratio of *EGFR* (IP/input) in OS compared to ST (*Figure 3C*). We also validated the significantly increased *EGFR* mRNA levels (*Figure 3D*) and decreased m⁶A enrichment in OS compared to ST by MeRIP-quantitative PCR (qPCR) (*Figure 3E*). To further test whether METTL3-mediated m⁶A regulates *EGFR* mRNA decay, we measured *EGFR* mRNA levels in ECs after treatment with the transcriptional inhibitor actinomycin D. Compared to the green fluorescent protein (GFP) control, METTL3 overexpression significantly decreased the remaining *EGFR* mRNA levels due to accelerated mRNA decay in the presence of functional m⁶A modification by overexpression of METTL3 (*Figure 3F*). As expected, *EGFR* mRNA showed a slower decay rate in response to METTL3 knockdown or OS treatment (*Figure 3G and H*). As we found that the 3'UTR of *EGFR* is a key region regulating m⁶A modification, we mutated two nearby potential m⁶A motifs AGACA and GGACT to AGTCA and GGTCT. Notably, the 3'UTR mutants did not respond to the decrease in luciferase activity caused by overexpression of METTL3 (*Figure 3I and J*). Collectively, these results suggest that OS-induced EC activation was mediated by m⁶A modification on *EGFR*.

## TSP-1/EGFR pathway participates in EC inflammation induced by METTL3 inhibition in response to OS

EGFR ligands that specifically activate EGFR include epidermal growth factor (EGF), transforming growth factor α (TGF-α), and TSP-1 (*Liu et al., 2009*). TSP-1 is a multidomain protein that contains EGF-like repeats that indirectly activate EGFR and selected downstream signaling pathways (*Garg et al., 2011*; *Liu et al., 2009*). TSP-1 is activated in shear-mediated arterial stiffening (*Kim et al., 2017*). To study the function of the endothelial TSP-1/EGFR axis in EC activation, we first treated wild-type and METTL3-overexpressed ECs with or without OS. Overexpression of METTL3 abolished phosphorylation of EGFR (Tyr-1068), AKT, and ERK, as well as total EGFR and VCAM-1 levels in response to OS (*Figure 4A and B*; *Figure 4—figure supplement 1A*). Furthermore, TSP-1 was significantly increased at both the mRNA and protein levels under OS (*Figure 4—figure supplement 1B*). Supplementation with recombinant human TSP-1 didn't change EGFR protein levels but enhanced EGFR phosphorylation which could be abolished by METTL3 overexpression (*Figure 4C and D*). Next, ECs were subjected to OS or METTL3 siRNA in the presence of the EGFR-selective tyrphostin AG1478, which blocked both OS and siMETTL3-mediated phosphorylation of EGFR, AKT, and ERK and VCAM-1 levels, had no effect on total EGFR expression (*Figure 4E–H*). The increased number of THP-1 (human myeloid leukemia mononuclear) cells adhering to HUVECs by OS or siMETTL3 was relieved in the presence of overexpression of METTL3 or AG1478 (*Figure 4F and H*; *Figure 4—figure supplement 1C-D*). These results suggest that METTL3 inhibits the transcriptional level of EGFR and downstream signaling events and cellular responses.

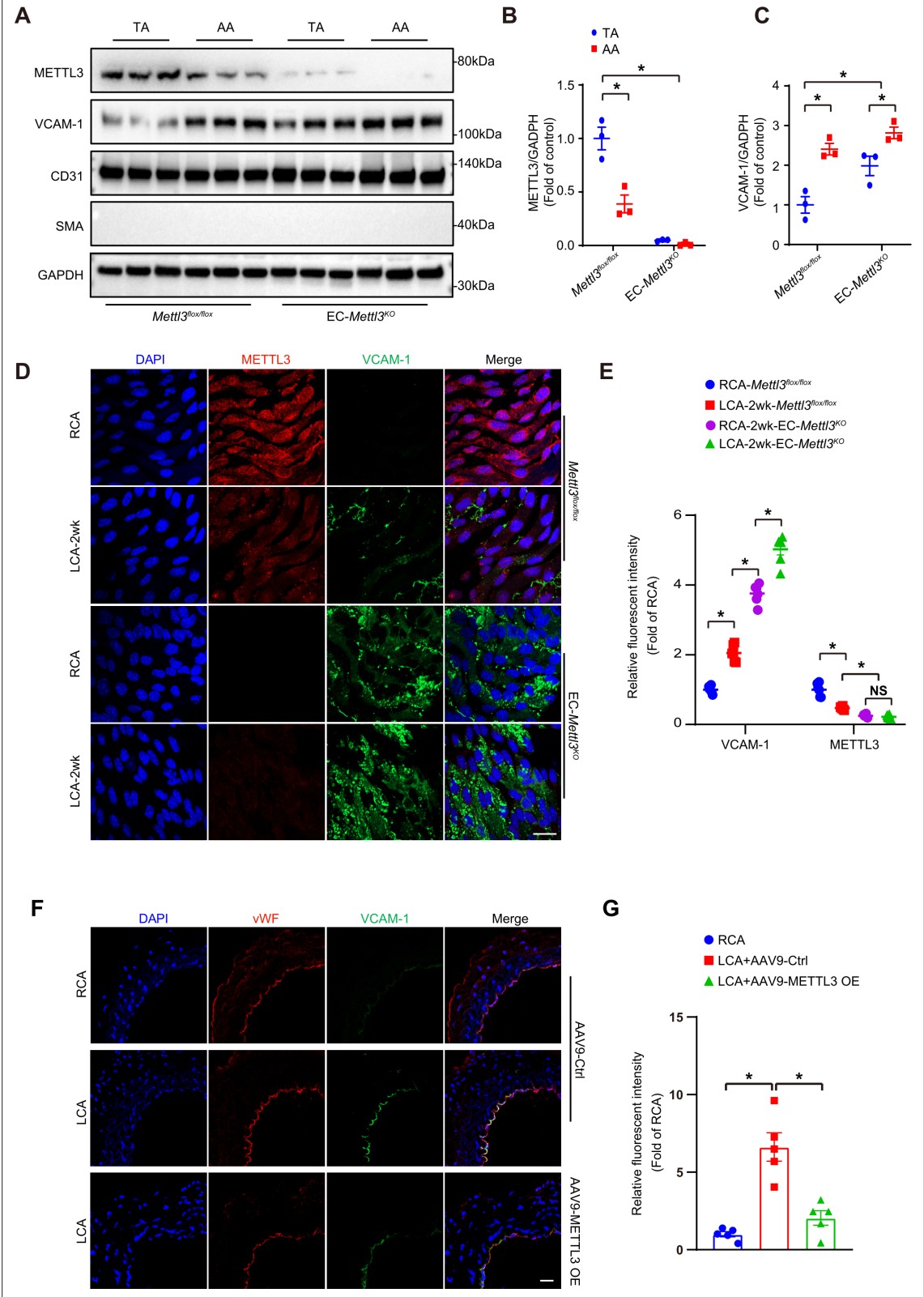

**Figure 2.** Methyltransferase like 3 (METTL3) deficiency induces endothelial activation in atheroprone regions. (**A–C**) Protein was extracted from the AA and TA of 8-week-old EC-*Mettl3*KO and *Mettl3*flox/flox mice. (**A**) Western blot analysis of the expression of METTL3, vascular adhesion molecule (VCAM-1), CD31, SMA (smooth muscle actin), and GAPDH in tissue lysates of AA and TA intima. AA, aortic arch; TA, thoracic aorta. (**B–C**) Quantification of protein expression in (**A**). Data are shown as the mean ± SEM, *p<0.05 (two-way ANOVA with Bonferroni multiple comparison post hoc test). Protein extracts

*Figure 2 continued on next page*

*Figure 2 continued*

of intima from three mice were pooled as one sample, n = 3. (**D**) EC-*Mettl3^KO* and *Mettl3^flox/flox* mice underwent partial ligation of the carotid artery for 2 weeks. En face immunofluorescence staining for the expression of VCAM-1 and METTL3 in ECs of the carotid artery of mice. Scale bar, 20 µm. (**E**) Quantification of the relative fluorescence intensity of VCAM-1 and METTL3. Data are shown as the mean ± SEM, *p<0.05 (two-way ANOVA with Bonferroni multiple comparison post hoc test). n = 6 mice. (**F**) Male mice underwent partial ligation of the carotid artery. During ligation, carotid arteries were infused with the indicated adeno-associated virus. Immunofluorescence staining of VCAM-1 and vWF in ECs of the RCA and LCA of mice. RCA, right carotid artery; LCA, left carotid artery. Scale bar, 20 µm. (**G**) Quantification of the relative fluorescence intensity of VCAM-1. Data are shown as the mean ± SEM, *p<0.05 (one-way ANOVA with Bonferroni multiple comparison post hoc test). n = 5 mice.

The online version of this article includes the following source data and figure supplement(s) for figure 2:

**Source data 1.** Mettl3 deficiency induces endothelial activation in atheroprone regions.

**Figure supplement 1.** Identification of EC-specific methyltransferase like 3 (*Mettl3*)-deficient mice.

**Figure supplement 1—source data 1.** Identification of EC-specific *Mettl3*-deficient mice.

## EC Mettl3 deficiency accelerates atherosclerosis in partial carotid artery ligated *Apoe^-/-* mice

To further verify the effect of EGFR and TSP-1 on endothelial function, we first evaluated EGFR and TSP-1 protein levels in partially ligated carotid arteries with or without AAV9-METTL3 OE infection and performed cross sections immunofluorescence staining after 2 weeks. Overexpression of METTL3 inhibited EGFR expression induced by EC Mettl3 deficiency in partially ligated carotid artery but had no effect on TSP-1 levels with partial ligation (*Figure 5A–D*; *Figure 2—figure supplement 1C* and D). The knockdown efficiency of METTL3 was approximately 95%, as confirmed by en face staining (*Figure 2A*; *Figure 5—figure supplement 1A* and B). Next, we performed partial ligation in *Apoe^-/-* EC-*Mettl3^KO* and *Apoe^-/-* *Mettl3^flox/flox* mice fed a Western-type diet (WTD) immediately after the surgery. Carotid Doppler ultrasonography verified stronger signals in the partially ligated LCA when fed a WTD compared to previous data in *Figure 1—figure supplement 1D* (*Figure 5—figure supplement 1C*). The lesion areas in the ligated carotid artery were more serious in *Apoe^-/-* EC-*Mettl3^KO* mice than in *Apoe^-/-* *Mettl3^flox/flox* mice at both 2 and 4 weeks (*Figure 5E and F*). The levels of plasma triglyceride and cholesterol and body weight did not change among the groups (*Figure 5—figure supplement 1D* and E). Thus, EC Mettl3 deficiency in partially carotid artery ligated mice triggers EGFR expression and EC activation.

## Mettl3-deficient ECs accelerates atherosclerosis in *Apoe^-/-* mice

To further detect the proatherogenic role of endothelial METTL3, we carried out a standard atherosclerosis study in *Apoe^-/-* EC-*Mettl3^KO* mice and *Apoe^-/-* *Mettl3^flox/flox* mice. After 12 weeks of WTD, Oil Red O staining of aortas revealed that, in comparison with *Apoe^-/-* *Mettl3^flox/flox* mice, Mettl3-deficient ECs significantly increased the total and AA atherosclerotic area in aortas (*Figure 6A and B*). Aortic root staining showed that Mettl3 deficiency increased the lesion area, lipid deposition, and macrophage infiltration, as well as EGFR and VCAM-1 expression, but had minimal effects on collagen fiber or vascular smooth muscle cell content (*Figure 6C–H*). These results indicate that METTL3 is an important effector in EC activation and atherogenesis.

## TSP1/EGFR signaling is involved in atherosclerosis

To investigate the function of the endothelial TSP1/EGFR axis in atherogenesis, the endothelium of *Apoe^-/-* EC-*Mettl3^KO* and *Apoe^-/-* *Mettl3^flox/flox* mice with a partially ligated carotid artery was infected with lentivirus-mediated *Thbs1* shRNA. The knockdown efficiency of *Thbs1* shRNA was confirmed by en face staining (*Figure 7—figure supplement 1A-B*). Two weeks after ligation, the lesion area in *Apoe^-/-* EC-*Mettl3^KO* mice was significantly increased by 2.5-fold compared to that in *Apoe^-/-* *Mettl3^flox/flox* mice. However, *Thbs1* knockdown successfully reversed the lesion area in both genotypes of mice (*Figure 7A and B*). The EGFR-selective tyrphostin AG1478 significantly reduced the phosphorylation of EGFR without affecting the total EGFR levels in both genotypes of mice (*Figure 7—figure supplement 1C-E*). Consistent with *Thbs1* knockdown, AG1478 exerted the same anti-atherosclerotic effect (*Figure 7A and B*). Both of them inhibited the expression of VCAM-1 induced by EC Mettl3 deficiency in partially ligated carotid artery (*Figure 7C and D*). The levels of plasma triglyceride and cholesterol

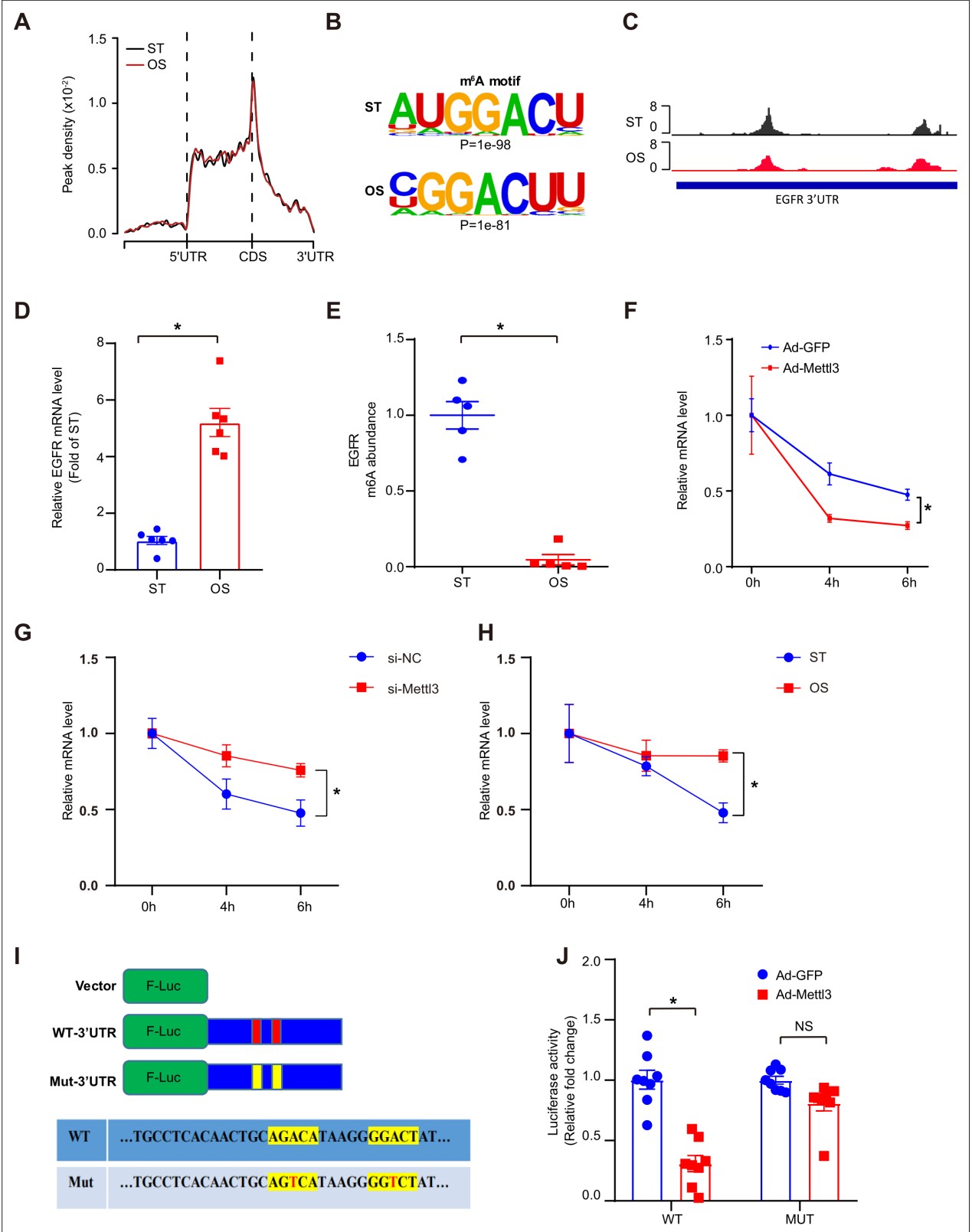

**Figure 3.** Oscillatory stress (OS)-abolished $N^6$-methyladenosine (m6A) prevents epidermal growth factor receptor (*EGFR*) mRNA degradation. (**A**) Distribution of m6A peaks along the 5' untranslated regions (5'UTR), CDS (coding sequence), and 3'UTR regions of mRNA in static treatment (ST) and OS. (**B**) m6A motif identified from human umbilical vein endothelial cells (HUVECs) under ST and OS treatments. (**C**) Integrative genomics viewer tracks displaying the results of IP *vs.* input read distributions in *EGFR* 3'UTR mRNA of HUVECs under ST and OS treatments. (**D**) qPCR analysis of *EGFR* mRNA

*Figure 3 continued on next page*

*Figure 3 continued*

levels in ST and OS. Data are shown as the mean ± SEM, *p<0.05 (Student's *t* test). n = 6. (**E**) MeRIP-qPCR detection of m$^6$A enrichment on *EGFR* mRNA in ST and OS. Data are shown as the mean ± SEM, *p<0.05 (Student's *t* test). n = 5. (**F–H**) qPCR analysis showing delayed *EGFR* mRNA degradation upon methyltransferase like 3 (*Mettl3*)-overexpression (**F**); si-*Mettl3* (**G**); and OS treatment (**H**). HUVECs were treated with actinomycin D for 4 and 6 hr. Data are shown as the mean ± SEM, *p<0.05 (two-way ANOVA with Bonferroni multiple comparison post hoc test). n = 6. (**I**) Schematic representation of the mutated (RRACH to RRTCH) 3' UTR of *EGFR* plasmids. (**J**) Relative activity of the wild-type or mutant *EGFR* 3'UTR firefly luciferase reporter in K293 cells treated with green fluorescent protein (GFP)- or GFP-*Mettl3*-overexpressing adenovirus. Data are shown as the mean ± SEM, *p<0.05, NS, not significant (Student's *t* test). n = 8. RNA-seq and MeRIP-seq data generated in this study have been deposited to the Genome Sequence Archive in BIG Data Center under accession number PRJCA004746.

The online version of this article includes the following source data and figure supplement(s) for figure 3:

**Source data 1.** OS-abolished m$^6$A prevents *EGFR* mRNA degradation.

**Figure supplement 1.** $N^6$-methyladenosine (m$^6$A) profiling in human umbilical vein endothelial cells treated with static treatment (ST) and oscillatory stress (OS).

were unchanged among the groups (*Figure 7—figure supplement 1F*). These data indicate that endothelial Mettl3-mediated m$^6$A in atheroprone regions is dependent upon TSP-1/EGFR in vivo.

## Discussion

Our results reveal that RNA m$^6$A modification plays an important role in regulating endothelial activation and the atherogenesis response to oscillatory flow. First, we found that m$^6$A modification levels were reduced concomitant with downregulation of METTL3 in atheroprone regions. Second, oscillatory flow-derived MeRIP-seq analysis revealed that m$^6$A modification of the *EGFR* 3'UTR contributes to atherogenesis. Third, overexpression of METTL3 restored m$^6$A modification and reversed VCAM-1 expression and the monocyte adhesion ability response to OS. Finally, TSP-1/EGFR inhibition prevented the development of atherosclerosis, suggesting a novel therapeutic method for atherosclerosis patients. Together, our findings demonstrate that METTL3 and m$^6$A modifications could alleviate endothelial activation and atherogenesis through accelerated degradation of oscillatory flow-induced *EGFR* mRNA expression.

Atherosclerotic plaques tend to develop in the vasculature locations with increased shear stress and OS. EC activation in response to oscillatory flow plays important roles in regulating circulatory functions and atherosclerosis development, which is involved in the expression of various genes (*Li et al., 2019*; *Zhou et al., 2014*). Recently, increasing evidence has suggested that m$^6$A mRNA modification participates in a number of biological functions and in progression of ECs (*Shi et al., 2019*; *Wang et al., 2014*; *Xiang et al., 2017*). In this study, we generated tamoxifen-inducible endothelium-specific Mettl3 deficiency mice with or without *Apoe*$^{-/-}$ background. Western blot analysis of arterial intima revealed decreased METTL3 in AA compared to TA in mice of both genotypes (*Figure 2A*). Overexpression of Mettl3 has potential protective effects on endothelial activation in atheroprone regions (*Figure 2F*), indicating the involvement of both m$^6$A modification and METTL3 in atherogenesis.

As the core methyltransferase subunit, METTL3 has been reported to modulate embryonic development (*Aguilo et al., 2015*) and spermatogenesis (*Xu et al., 2017*), and its deletion in mice causes early embryonic lethality (*Geula et al., 2015*). METTL3 also plays a central role in osteogenic differentiation and inflammatory response (*Wu et al., 2018*; *Zhang et al., 2019*), which act similarly in ECs. METTL3 expression is upregulated during osteoblast differentiation and downregulated after LPS (lipopolysaccharide) stimulation, while its depletion enhances the expression of *Smad* genes and proinflammatory cytokine expression in MAPK and NF-κB signaling pathways (*Zhang et al., 2019*). Chien et al. proposed that METTL3 is increased after long term of disturbed flow for 48 hr (*Chien et al., 2021*). We performed a short term of disturbed flow for 6 hr, and the mechanism research in genotype mice limited in 2 or 4 weeks. Furthermore, most evidence was demonstrated in EC-specific Mettl3 knockout mice. We presume a possibility that the differential expression of METTL3 is attributed to variations in the stimulus, such as treatment time and strength. The development of hematopoietic stem cells (HSPCs) also requires METTL3-mediated m$^6$A modification, and mettl3 deficiency in arterial endothelial cells blocks endothelial-to-hematopoietic transition, followed by suppression of the

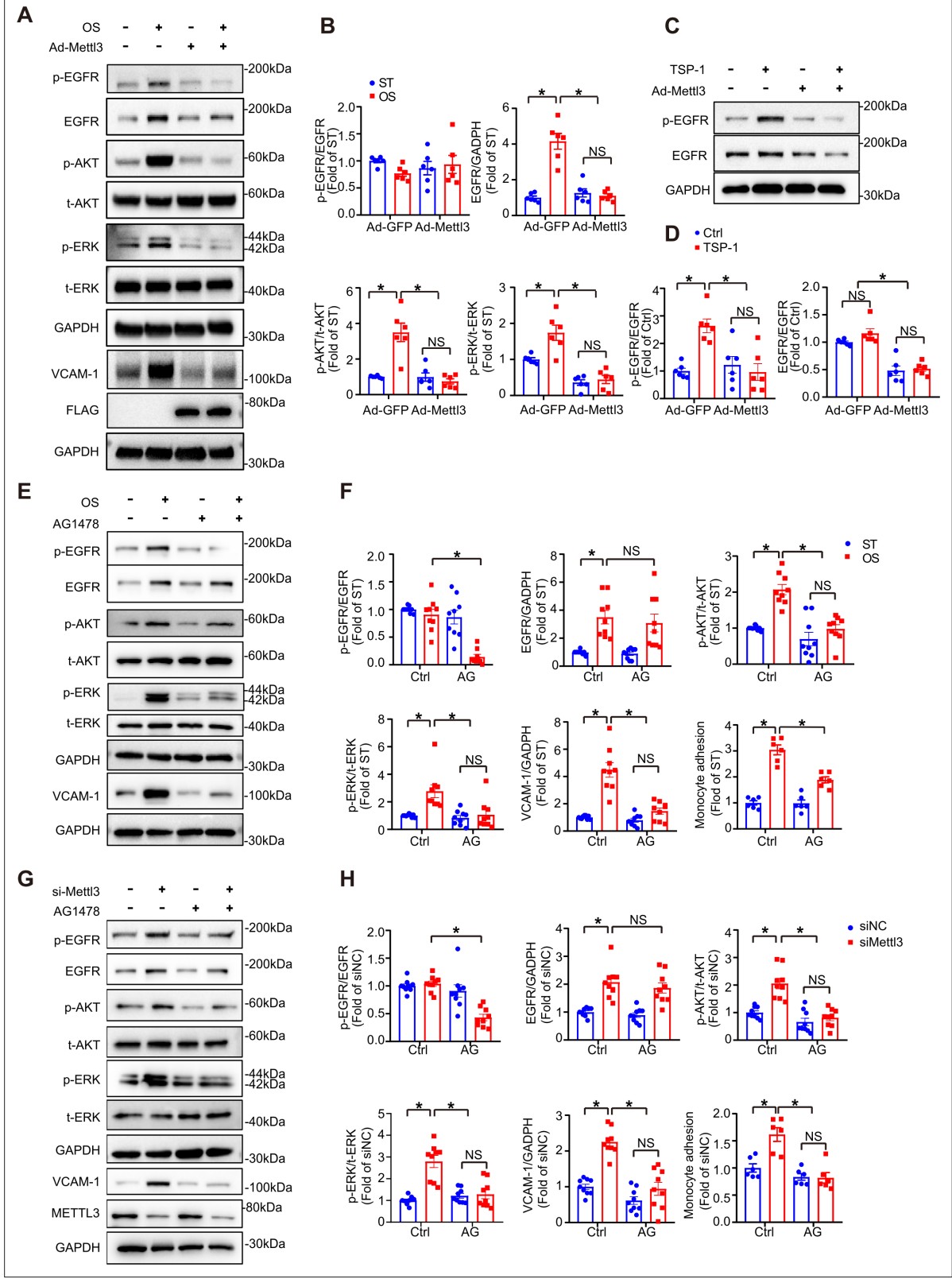

**Figure 4.** The thrombospondin-1/epidermal growth factor receptor (TSP-1/EGFR) pathway participates in EC inflammation induced by methyltransferase like 3 (METTL3) inhibition in response to oscillatory stress (OS). (**A**) Western blot analysis of p-EGFR, EGFR, p-AKT, t-AKT, p-ERK, t-ERK, FLAG (tag of METTL3), and vascular adhesion molecule 1 (VCAM-1) expression. GAPDH is the protein loading control. Human umbilical vein endothelial cells (HUVECs) were infected with the indicated adenoviruses for 24 hr with or without exposure to OS or static treatment (ST) for another

*Figure 4 continued on next page*

*Figure 4 continued*

6 or 12 hr. (**B**) Quantification of the expression of the indicated proteins in (**A**). Data are shown as the mean ± SEM, *p<0.05, NS, not significant (two-way ANOVA with Bonferroni multiple comparison post hoc test). n = 6. (**C**) HUVECs were infected with the indicated adenoviruses for 24 hr with or without TSP-1 (10 µg/ml) treatment. Western blot analysis of p-EGFR, EGFR, and GAPDH. (**D**) Quantification of the expression of the indicated proteins in (**C**). Data are shown as the mean ± SEM, *p<0.05, NS, not significant (two-way ANOVA with Bonferroni multiple comparison post hoc test). n = 6. (**E**) HUVECs were exposed to OS or ST for 6 or 12 hr with or without pretreatment with AG1478 (10 µmol/L). Western blot analysis of p-EGFR, EGFR, p-AKT, t-AKT, p-ERK, t-ERK, VCAM-1, and GAPDH. (**F**) Quantification of the expression of the indicated proteins in (**E**). Data are shown as the mean ± SEM, *p<0.05, NS, not significant (two-way ANOVA with Bonferroni multiple comparison post hoc test). n = 9. (**G**) HUVECs were infected with METTL3 siRNA for 24 hr with or without treatment with AG1478 (10 µmol/L). Western blot analysis of p-EGFR, EGFR, p-AKT, t-AKT, p-ERK, t-ERK, VCAM-1, METTL3, and GAPDH. (**H**) Quantification of the expression of the indicated proteins in (**G**). Data are shown as the mean ± SEM, *p<0.05, NS, not significant (two-way ANOVA with Bonferroni multiple comparison posttest). n = 9. THP-1 cells were labeled with fluorescence dye, and then a cell adhesion assay was performed. The number of adherent cells was normalized to that of HUVECs as a control (statistical chart in **F**, **H**). Data are shown as the mean ± SEM, *p<0.05, NS, not significant (two-way ANOVA with Bonferroni multiple comparison post hoc test). n = 6.

The online version of this article includes the following source data and figure supplement(s) for figure 4:

**Source data 1.** The TSP-1/EGFR pathway participates in EC inflammation in response to OS.

**Figure supplement 1.** Thrombospondin-1 (TSP-1) and epidermal growth factor receptor (EGFR) are involved in methyltransferase like 3 (METTL3)-mediated EC dysfunction in response to oscillatory stress (OS).

**Figure supplement 1—source data 1.** TSP-1 and EGFR are involved in METTL3-mediated EC dysfunction in response to OS.

generation of HSPCs through activation of Notch signaling (*Zhang et al., 2017*). METTL3 plays a similar role in cerebral arteriovenous malformation, and deletion of METTL3 in ECs significantly affects angiogenesis by reducing heterodimeric Notch E3 ubiquitin ligase formed by DTX1 and DTX3L (*Wang et al., 2020*). METTL3 seems to be essential for maintaining endothelial function. Here, we analyzed RNA-seq and MeRIP-seq to reveal declined methylation and increased expression of cell adhesion- and migration-related pathways. Our results describe the role of m⁶A and METTL3 in atherogenic progression and identify therapeutic strategies against atherosclerosis through m⁶A modification and its related targets.

It has been reported that EGFR plays a role in foam cell transformation and accelerates atherosclerotic lesions characterized by accumulation of smooth muscle cells and macrophages (*Wang et al., 2017*). Although EGFR is expressed at low levels in ECs, explosive elevation of transcription and phosphorylation of EGFR were detected after OS treatment (*Figure 3D* and *Figure 4*), consistent with a report from *Rizvi et al., 2013*. METTL3 binds to *EGFR* mRNA near the stop codon and is responsible for its m⁶A modification, modulating *EGFR* mRNA stability in human cancer cells (*Lin et al., 2016*). Chien et al. also pointed out EGFR as a hypomethylated gene in METTL3 knockdown under OS (*Chien et al., 2021*). Our results demonstrated that the *EGFR* 3'UTR response to OS and double mutation of the 3'UTR abolished the reduction induced by METTL3 overexpression in ECs (*Figure 3*), indicating that EGFR is responsible for m⁶A modification in response to OS treatment. EGFR is activated in response to its EGF-like repeats, such as TSP1, which require matrix metalloprotease activity, and the ligand-binding portion of the EGFR ectodomain (*Liu et al., 2009*). The EGFR signaling pathway was blocked by overexpression of METTL3 in response to OS and supplementation with TSP-1 (*Figure 4A–D*). However, TSP-1 was not affected by METTL3 intervention, and we presume that TSP-1 acts as a specific ligand that activates EGFR after OS treatment. Moreover, the EGFR selective tyrphostin AG1478 inhibited EGFR phosphorylation at tyrosine 1068 without affecting the total levels of EGFR (*Garg et al., 2011*; *Wang et al., 2017*) and had an atheroprotective role (*Wang et al., 2017*). AG1478 reduced levels of p-EGFR and downstream proteins in the presence of OS, and *Thbs1* shRNA and AG1478 reversed atherogenesis in partially ligated arteries in EC-Mettl3^KO mice (*Figure 4* and *Figure 7*).

In summary, we provide both in vitro and in vivo evidence demonstrating that m⁶A RNA modification regulates progression of EC activation in response to the onset of OS and subsequently, early atherogenesis. It should be noted that METTL3-modified *EGFR* mRNA participates in the pathogenic

mechanism of atherogenesis. The TSP-1/EGFR axis may contribute to m$^6$A-modified atherogenesis, and inhibition of the axis helps to retard atherosclerosis.

# Materials and methods

## Key resources table

| Reagent type (species) or resource | Designation | Source or reference | Identifiers | Additional information |
|---|---|---|---|---|
| Antibody | anti-METTL3 (Rabbit monoclonal) | Cell Signaling Technology | Cat# 96,391 | WB (1:1000) |
| Antibody | anti-METTL14 (Rabbit monoclonal) | Cell Signaling Technology | Cat# 51,104 | WB (1:1000) |
| Antibody | anti-METTL16 (Rabbit monoclonal) | Cell Signaling Technology | Cat# 17,676 | WB (1:1000) |
| Antibody | anti-WTAP (Rabbit monoclonal) | Cell Signaling Technology | Cat# 56,501 | WB (1:1000) |
| Antibody | anti-Virillizer (Rabbit monoclonal) | Cell Signaling Technology | Cat# 88,358 | WB (1:1000) |
| Antibody | anti-phospho-EGFR (Rabbit monoclonal) | Cell Signaling Technology | Cat# 3,777 | WB (1:1000) |
| Antibody | anti-EGFR (Rabbit monoclonal) | Cell Signaling Technology | Cat# 4,267 | WB (1:1000) |
| Antibody | anti-VCAM-1 (Rabbit monoclonal) | Cell Signaling Technology | Cat# 15,631 | WB (1:1000) |
| Antibody | anti-VCAM-1 (Rabbit monoclonal) | Cell Signaling Technology | Cat# 39,036 | IF (1:100) |
| Antibody | anti-thrombospondin-1 (Rabbit monoclonal) | Cell Signaling Technology | Cat# 37,879 | WB (1:1000) |
| Antibody | anti-αSMA (Rabbit monoclonal) | Cell Signaling Technology | Cat# 19,245 | WB (1:1000) |
| Antibody | anti-phospho-ERK (Rabbit monoclonal) | Cell Signaling Technology | Cat# 8,544 | WB (1:1000) |
| Antibody | anti-phospho-AKT (Rabbit monoclonal) | Cell Signaling Technology | Cat# 5,012 | WB (1:1000) |
| Antibody | anti-ERK (Mouse monoclonal) | Santa Cruz | Cat# sc-271269 | WB (1:1000) |
| Antibody | anti-AKT (Mouse monoclonal) | Santa Cruz | Cat# sc-5298 | WB (1:1000) |
| Antibody | anti- METTL3 (Rabbit monoclonal) | Proteintech | Cat# 15073–1-AP | IF (1:100) |
| Antibody | anti- GFP (Rabbit monoclonal) | Proteintech | Cat# 50430–2-AP | IF (1:100) |
| Antibody | anti- GAPDH (Rabbit monoclonal) | Proteintech | Cat# 60004–1-Ig | WB (1:5000) |
| Antibody | anti- EGFR (Rabbit monoclonal) | Abcam | Cat# ab52894 | IF (1:100) |
| Antibody | anti- VE-cadherin (Rat monoclonal) | Abcam | Cat# ab33168 | IF (1:100) |
| Antibody | anti- CD31 (Rabbit monoclonal) | Abcam | Cat# ab24590 | IF (1:100) |
| Antibody | anti- CD68 (Rabbit monoclonal) | Abcam | Cat# ab955 | IF (1:100) |
| Antibody | anti- vWF (Sheep monoclonal) | Abcam | Cat# ab11713 | IF (1:100) |
| Antibody | anti-thrombospondin-1 (Mouse monoclonal) | Abcam | Cat# ab1823 | IF (1:100) |
| Antibody | anti- thrombospondin-1 (Mouse monoclonal) | Abcam | Cat# ab1823 | IF (1:100) |

*Continued on next page*

*Continued*

| Reagent type (species) or resource | Designation | Source or reference | Identifiers | Additional information |
|---|---|---|---|---|
| Antibody | Alex 488-conjugated goat anti-rabbit antibody | Thermo Fisher Scientific | Cat# A-11008 | IF (1:200) |
| Antibody | Alex 594-conjugated goat anti-mouse antibody | Thermo Fisher Scientific | Cat# A-11008 | IF (1:200) |
| Antibody | Alex 488-conjugated goat anti-rabbit antibody | Thermo Fisher Scientific | Cat# A-11005 | IF (1:200) |
| Antibody | Alex 594-conjugated donkey anti-sheep antibody | Thermo Fisher Scientific | Cat# A-11016 | IF (1:200) |
| Chemical compound, drug | AG1478 | Selleck | Cat# S2728 | |
| Chemical compound, drug | Recombinant Human Thrombospondin-1 | Absin | Cat# abs 04665 | 1.03 mg/ml |
| sequence-based reagent | Human EGFR-3utr | This paper | N/A | Sequences in *Supplementary file 3* |
| sequence-based reagent | Human EGFR | This paper | N/A | Sequences in *Supplementary file 3* |
| sequence-based reagent | Human THBS1 | This paper | N/A | Sequences in *Supplementary file 3* |
| sequence-based reagent | Human GAPDH | This paper | N/A | Sequences in *Supplementary file 3* |
| software, algorithm | Ingenuity Pathway Analysis | National Clinical Research Center for Blood Diseases | http://www.ingenuity.com/ | |
| software, algorithm | Prism version 8.0 | GraphPad Software Inc | https://www.graphpad.com/scientific-software/prism/ | |

## Cell Culture and Shear Stress Experiments

HUVECs were isolated and cultured as described (*He et al., 2018*; *Li et al., 2019*). Mouse aortic endothelial cells (catalog no. CP-M075) and specific medium (CM-M075) were purchased from Procell (Wuhan, China). For flow experiments, confluent monolayers of HUVECs were seeded on glass slides, and a parallel plate flow system was used to impose oscillatory flow ($0.5 \pm 4$ dyn/cm$^2$). THP-1 cells (ATCC, catalog TIB-202) were cultured with 1640 medium supplemented with 10% fetal bovine serum. The flow system was enclosed in a chamber held at 37°C and ventilated with 95% humidified air plus 5% $CO_2$.

## Recombinant Mettl3 adenovirus, adeno-associated virus, and THBS1 lentivirus construction

Adenoviruses expressing green fluorescent protein and Ad-Flag-tagged human METTL3 (Ad-METTL3) were purchased from GeneChem (Shanghai, China). Recombinant AAV serotype nine vectors carrying METTL3 or empty vector were manufactured by GeneChem Co, Ltd (Shanghai, China). Lentiviruses carrying short hairpin RNA (shRNA) targeting *Thbs1* (LV-sh*Thbs1*) and nonspecific shRNA (LV-sh*Ctrl*) were constructed by Shanghai Genechem Co (Shanghai). HUVECs were infected with adenovirus at a multiplicity of infection (MOI) of 10, and no detectable cellular toxicity was observed.

## Western blot analysis

Cells or tissues were homogenized in cold RIPA lysis buffer supplemented with complete protease inhibitor cocktail and phosSTOP phosphatase inhibitor (Roche). Proteins were resolved by SDS-PAGE and transferred to NC membranes (Bio-Rad). Target proteins were detected using specific primary antibodies (1:1000). Bound antibodies were detected by horseradish peroxidase-conjugated secondary antibody (1:5000) and visualized by enhanced chemiluminescence (Cell Signaling Technology).

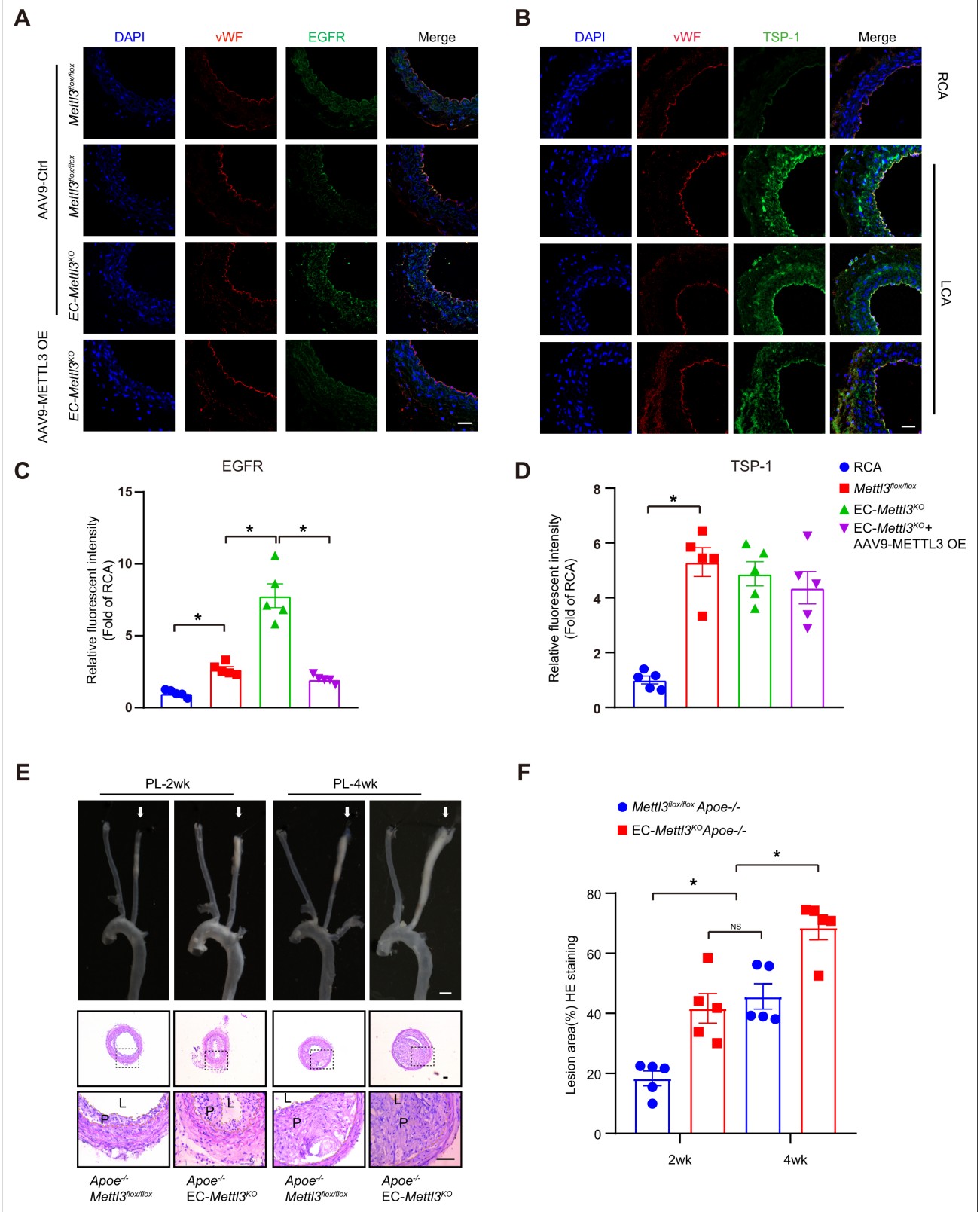

**Figure 5.** Epidermal growth factor receptor (EGFR) contributes to EC activation in endothelial methyltransferase like 3 (Mettl3)-deficient mice.
(**A–B**) EC-*Mettl3*[KO] and *Mettl3*[flox/flox] mice underwent partial ligation of the carotid artery for 2 weeks were infused with the indicated adeno-associated virus. Immunofluorescence staining for expression of EGFR, thrombospondin-1 (TSP-1) in ECs of the carotid artery of mice. Scale bar, 80 μm.
(**C–D**) Quantification of the relative fluorescence intensity of EGFR and TSP-1. Data are shown as the mean ± SEM, *p<0.05 (two-way ANOVA with

*Figure 5 continued on next page*

*Figure 5 continued*

Bonferroni multiple comparison post hoc test). n = 5 mice. (**E**) Eight-week-old male *Apoe*$^{-/-}$*Mettl3*$^{flox/flox}$ and *Apoe*$^{-/-}$ EC-*Mettl3*$^{KO}$ mice with 2 or 4 weeks of partial ligation were fed a Western-type diet, and arterial tissues were isolated to examine atherosclerotic lesions. Scale bar: 1.5 mm. Ligated coronary arteries were sectioned for hematoxylin-eosin staining. Scale bar: 100 µm. L, lumen; P, plaque. (**F**) Quantification of lesion area. Data are shown as the mean ± SEM, *p<0.05, NS, not significant (one-way ANOVA with Bonferroni multiple comparison post hoc test). n = 5 mice.

The online version of this article includes the following source data and figure supplement(s) for figure 5:

**Source data 1.** EGFR contributes to EC activation in endothelial Mettl3-deficient mice.

**Figure supplement 1.** Verification of disturbed flow in the partially ligated left common carotid artery.

**Figure supplement 1—source data 1.** Verification of disturbed flow in the partially ligated left common carotid artery.

## Cell adhesion assay

THP-1 cells were labeled with CellTrace calcein red-orange AM (Thermo Fisher catalog C34851) and then plated onto HUVEC plates at $2 \times 10^6$ cells/well. After incubation for 60 min at 37°C, nonadherent cells were removed by three washes with phosphate buffered saline (PBS). The numbers of stained adherent cells in five random fields were counted for each group under a fluorescence microscope.

## Immunofluorescence staining

Arteries or aorta sections were fixed with 4% paraformaldehyde for 15 min. After permeabilization/blocking in 0.05% Triton X-100 (in PBS) and 1% bovine serum albumin (BSA) for 30 min at room temperature, aortas were incubated at 4°C overnight in incubation buffer containing 1% BSA and primary antibodies (1:100) against METTL3, p-EGFR, EGFR, VCAM-1, GFP, αSMA, CD68, vWF, TSP-1, and CD31. After washing in PBS three times, aortas were incubated with Alexa Fluor 488- or Alexa Fluor 594-conjugated secondary antibodies (1:200) for 1 hr at room temperature. Fluorescent signals were detected by using a Zeiss confocal laser scanning microscope.

## Animals

We established tamoxifen-inducible EC-specific Mettl3-deficient (EC-*Mettl3*$^{KO}$) and littermate control (*Mettl3*$^{flox/flox}$) mice. Mice carrying the floxed *Mettl3* allele mice were crossed with mice harboring Cre recombinase under the control of the Cdh5 promoter, which contained a tamoxifen-inducible EC-specific Cre. Tamoxifen was administered once every 24 hr for five consecutive days. All mice were on a C57BL/6 J background and were maintained under a 12:12 hr light/dark cycle (lights on at 7:00 and lights off at 19:00). The investigation conformed to the Guide for the Care and Use of Laboratory Animals by the US National Institutes of Health (NIH Publication No. 85–23, revised in 2011). All study protocols and the use of animals were approved by the Institutional Animal Care and Use Committee of Tianjin Medical University.

## Animal experiments

Partial ligation of carotid artery was performed as described (*Zhang et al., 2020*). The indicated mice were anesthetized by using isoflurane (2–3%). Carotid arteries of both sides were exposed by creating a ventral midline incision (4–5 mm) in the neck. The left external carotid, internal carotid, and occipital arteries were ligated and right external carotid as sham; the superior thyroid artery was left intact. Mice were monitored until recovery in a chamber on a heating pad after surgery. Mice were fed WTD for 2 and 4 weeks. For lentivirus infection studies, a single exposure of $5 \times 10^5$ TU adenovirus was lumenally delivered to the left carotid artery and kept inside for 40 min to allow for sufficient infection. The lentivirus solution was subsequently removed and blood flow was restored. For AAV9 infection studies, AAV9-TIEp-METTL3/empty vectors ($3 \times 10^{11}$ vector genomes/mice) were delivered by intravenous injection. One week after AAV9 delivery, partial ligation was performed. Mice were fed WTD immediately after the surgery for 2 weeks.

## Quantification of lipid levels

Blood samples were collected by tail bleeding into heparin-coated tubes. Plasma was separated by centrifugation. Total plasma cholesterol, triglycerides, low-density lipoprotein-cholesterol, and high-density lipoprotein-cholesterol levels were measured by using kits from BioSino Bio-Technology and Science Inc (Beijing, China).

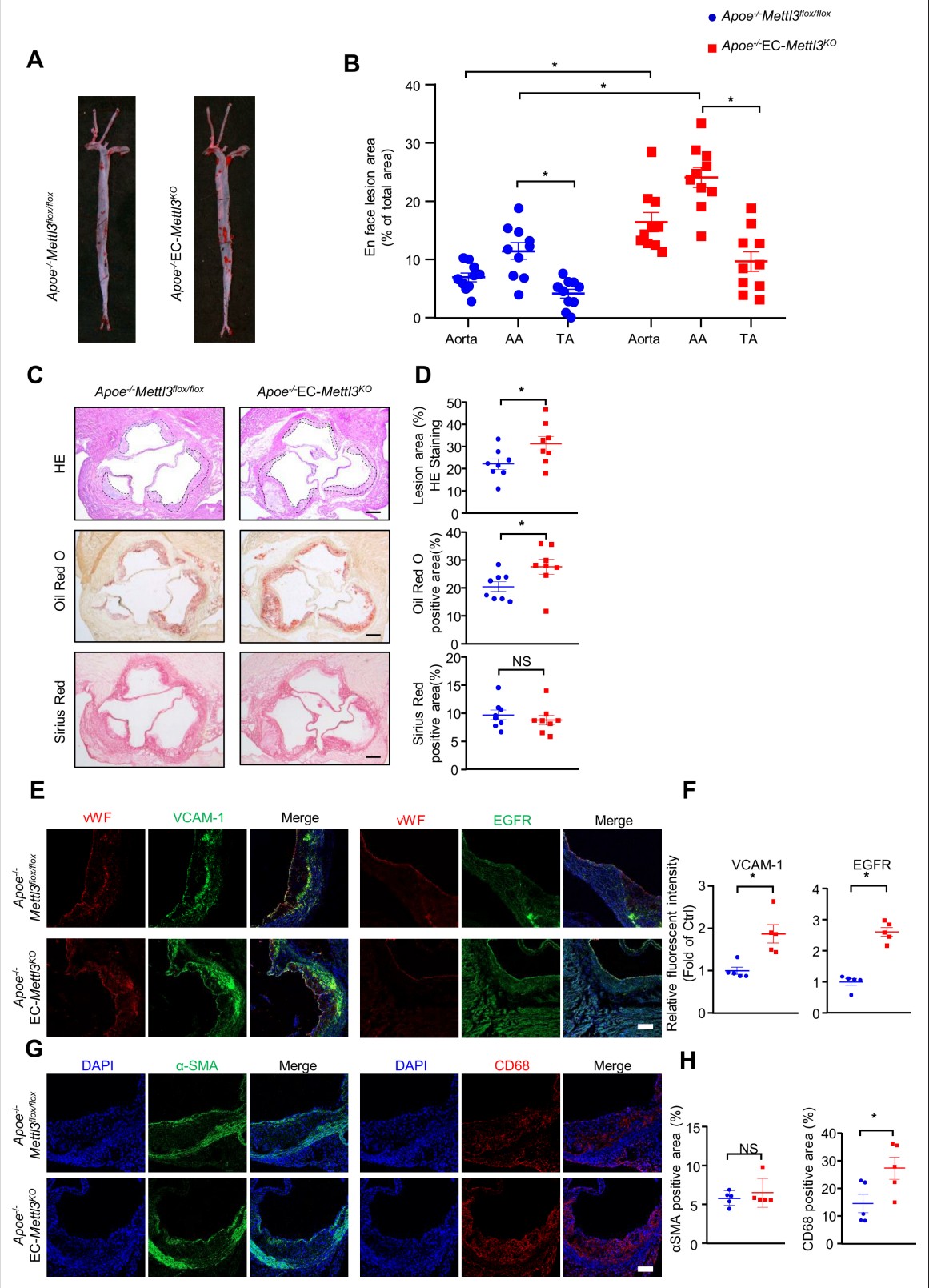

**Figure 6.** EC-specific methyltransferase like 3 (METTL3) deficiency accelerates atherogenesis in *Apoe*[-/-] mice. *Apoe*[-/-]EC-*Mettl3*[KO] and *Apoe*[-/-]*Mettl3*[flox/flox] mice were fed a Western-type diet for 12 weeks. (**A**) Oil Red O staining of aortas. (**B**) Plaque area as a percentage of total area. AA, aortic arch; TA, thoracic aorta. Data are shown as the mean ± SEM, *p<0.05 (two-way ANOVA with Bonferroni multiple comparison post hoc test). n = 10. (**C–D**) HE, Oil Red O, and Sirius Red immunofluorescence staining of aortic roots. White dashed line indicates the size of plaque. Quantification of plaque size, Oil Red

*Figure 6 continued on next page*

*Figure 6 continued*

O-positive area in plaque size, Oil Red O-positive area in plaque and collagen fiber (Sirius Red). Data are shown as the mean ± SEM, *p<0.05 (Student's *t* test). n = 8. (**E and G**) Epidermal growth factor receptor (EGFR), vascular adhesion molecule 1 (VCAM-1), vWF, α-SMA, and CD68 immunofluorescence staining of aortic roots. Scale bar, 20 µm. (**F and H**) Quantification of the relative fluorescence intensity of VCAM-1, EGFR, α-SMA, and CD68. Data are shown as the mean ± SEM, *p<0.05, NS, not significant (Student's *t* test). n = 5.

The online version of this article includes the following source data for figure 6:

**Source data 1.** EC-specific METTL3 deficiency accelerates atherogenesis in *Apoe*[-/-] mice.

## Total RNA extraction and real-time quantitative PCR analysis

Total RNA was extracted from tissue and HUVECs using RNA extraction kits (TransGen Biotech, ER501-01, China). Complementary DNA was synthesized using reverse transcription using SuperScript III and random primers (Thermo Fisher, catalog 12574035, MA). Real-time qPCR was performed using the Brilliant II SYBR Green qPCR Master Mix (Stratagene, CA) and the ABI 7900HT Real-Time PCR System (Life Technologies, CA). The results were normalized to those of 18 S. Data were calculated using comparative Ct values. The primer sequences are listed in *Supplementary file 3*.

## RNA purification and mRNA purification

Total RNA was extracted using TRIzol (Thermo Fisher), treated with Turbo DNase (Thermo Fisher), and then subjected to mRNA purification with the Dynabeads protein A mRNA purification kit (Thermo Fisher) following the manufacturer's instructions.

## UHPLC-MRM-MS/MS analysis of mononucleotides

An amount of 200 ng purified mRNA of each sample was digested with 0.1 U Nuclease P1 (Sigma, catalog N863) and 1.0 U calf intestinal phosphatase (New England Biolabs, catalog M0290), in the final reaction volume of 50 µl and incubated at 37°C for over 5 hr. The mixture was then filtered with ultra-filtration tubes (MW cutoff: 3 KDa, Pall, Port Washington, NY) through centrifuging at 14,000 g for 20 min. The samples were subjected to UHPLC-MRM-MS/MS analysis for detection of m⁶A and rA. The UHPLC-MRM-MS/MS analysis was performed according to the previous report (*Xiao et al., 2016*).

## RNA-seq

RNA-seq libraries were directly constructed using the KAPA RNA HyperPrep Kit (KAPA Biosystems) following the manufacturer's instructions. Sequencing was performed on an MGISEQ-2000 platform with a single end 50 nt read length. Three replicates were used in RNA-seq studies.

## MeRIP-seq

MeRIP was performed according to previously described methods. First, 5 µg of anti-m⁶A antibody (Synaptic Systems, 202003) was incubated with 20 µL Dynabeads Protein A (Invitrogen, 1,001D) in 500 µL IPP (immunoprecipitation buffer) buffer (150 mM NaCl, 10 mM Tris-HCl, pH 7.4, 0.1% NP-40, 10 U RNase Inhibitor) at 4°C for 1 hr. Second, ~ 100 ng of fragmented mRNA was added to the antibody-bead mixture and incubated at 4°C for 4 hr with gentle rotation. After extensive washing with IPP buffer, high-salt wash buffer (500 mM NaCl, 10 mM Tris-HCl, pH 7.4, 0.1% NP-40), and low-salt wash buffer (50 mM NaCl, 10 mM Tris-HCl, pH 7.4, 0.1% NP-40), RNA fragments were eluted from the beads with proteinase K (Roche, 3115836001) digestion at 55°C for 1 hr and extracted with phenol-chloroform extraction and ethanol precipitation. The recovered RNAs were subjected to library preparation using the KAPA RNA HyperPrep kit (KAPA Biosystems, KK8541). Sequencing was performed on an Illumina HiSeq X-Ten platform with paired end 150 base pair (bp) read length.

## Sequencing data analysis

For general preprocessing, SOAPnuke (version 1.5.2) and Trim Galore (version 0.6.4) were used to trim off the adapter sequences and low-quality bases for all samples. The remaining reads were aligned to the human genome (version hg19) using Hisat2 (version 2.0.5) (*Kim et al., 2015*), and only uniquely mapped reads with q ≥ 20 were used in subsequent analysis.

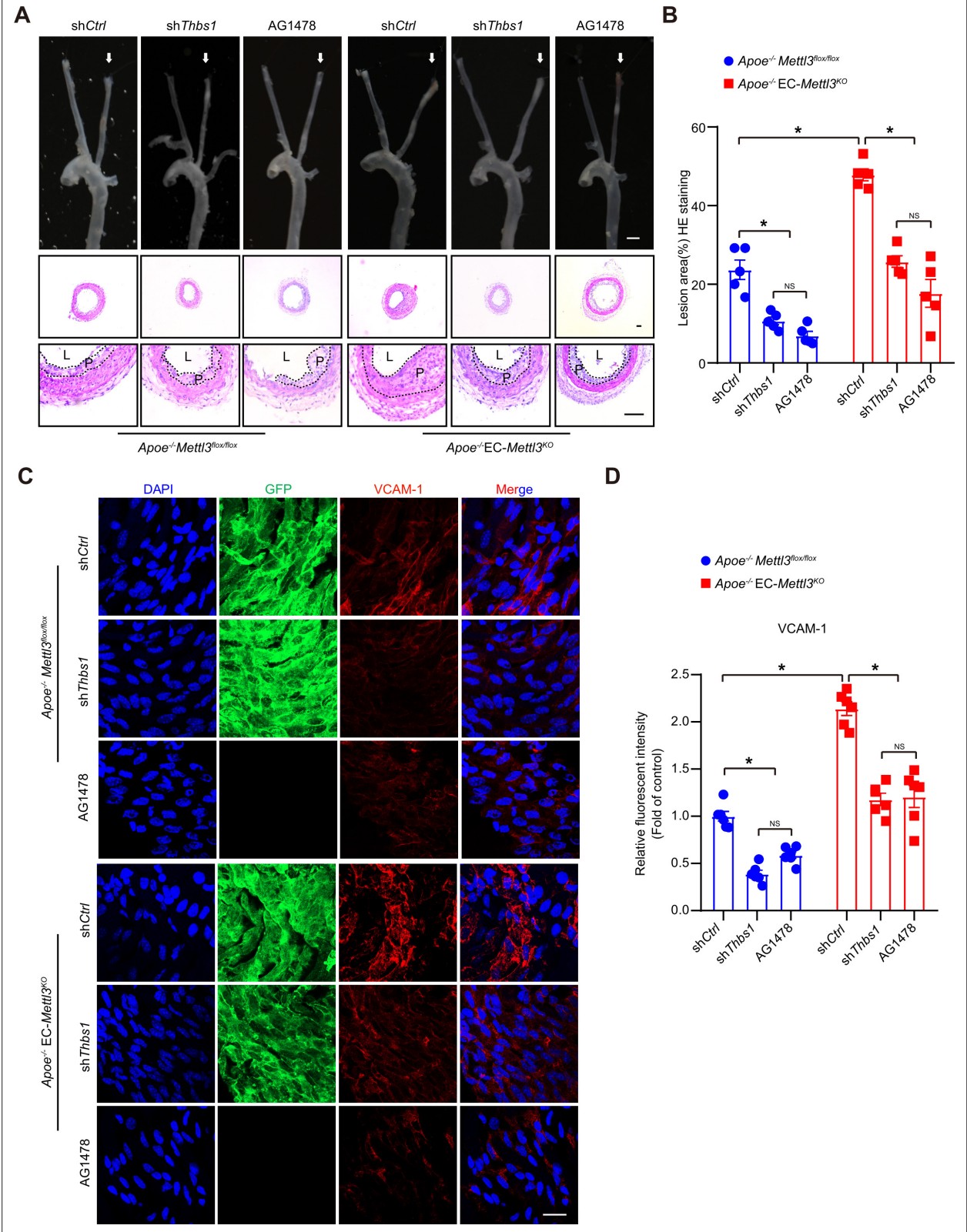

**Figure 7.** Thrombospondin-1/epidermal growth factor receptor (TSP1/EGFR) signaling is involved in atherosclerosis. (**A**) An 8-week-old male *Apoe$^{-/-}$ Mettl3$^{flox/flox}$* and *Apoe$^{-/-}$* EC-*Mettl3$^{KO}$* mice with 2 weeks of partial ligation were infused with the indicated lentiviruses or pretreated with AG1478 (AG, 10 mg/kg/day) for 7 days, and arterial tissues were isolated to examine atherosclerotic lesions. Scale bar: 1.5 mm. Ligated coronary arteries (LCAs) were sectioned for hematoxylin-eosin (H&E) staining. Scale bar: 100 µm. L, lumen; P, plaque. (**B**) Quantification of lesion area. Data are shown as the mean ±

*Figure 7 continued on next page*

*Figure 7 continued*

SEM, *p<0.05, NS, not significant (two-way ANOVA with Bonferroni multiple comparison post hoc test). n = 5 mice. (**C**) *Apoe*$^{-/-}$ *Mettl3*$^{flox/flox}$ and *Apoe*$^{-/-}$ EC-*Mettl3*$^{KO}$ mice with 2 weeks of partial ligation were infused with the indicated lentiviruses or pretreated with AG1478 (AG, 10 mg/kg/day) for 7 days. En face immunofluorescence staining of the expression of vascular adhesion molecule (VCAM-1) in ECs of the carotid artery of mice. Scale bar, 20 µm. (**D**) Quantification of the relative fluorescence intensity of VCAM-1. Data are shown as the mean ± SEM, *p<0.05, NS, not significant (two-way ANOVA with Bonferroni multiple comparison post hoc test). n = 6.

The online version of this article includes the following source data and figure supplement(s) for figure 7:

**Source data 1.** TSP1/EGFR signaling is involved in atherosclerosis.

**Figure supplement 1.** AG1478 decreases phosphorylation of epidermal growth factor receptor (EGFR) induced by oscillatory stress.

**Figure supplement 1—source data 1.** AG1478 decreases phosphorylation of EGFR induced by OS.

For MeRIP-seq, two biological replicates were conducted. The replicates of each sample were merged for m⁶A peak calling using MACS2 (version 2.1.4) (*Zhang et al., 2008*) with the corresponding input samples as a control. Default parameters were used, except for '-nomodel –keep dup all', to turn off fragment size estimation and to keep all uniquely mapped reads in MACS2. Finally, each peak was annotated based on Ensembl (release 72) gene annotation information using BETools intersectBed (version 2.28.0) (*Quinlan and Hall, 2010*). For RNA-seq, the number of reads mapped to each gene (Ensembl 72) was counted using featureCounts (version 1.6.2) (*Liao et al., 2014*) with the default parameters, except for '-s 2'.

## Statistical analysis of differentially expressed genes and gene ontology analysis

Differentially expressed genes between ST and OS treatments were identified using DEGseq (*Wang et al., 2010*). Differentially expressed genes were identified by $\log_2$ |fold change| > $\log_2$ |1.5| and FDR < 0.05. GO analysis was performed using DAVID (https://david.ncifcrf.gov/), and p < 0.05 were considered significantly enriched.

## Motif identification within m⁶A peaks and differential m⁶A peaks

HOMER (version 4.7) (*Heinz et al., 2010*) was used to identify the motif enriched by the m⁶A peak, and the motif length was limited to seven nt. Peaks annotated to mRNA were considered target sequences, and background sequences were constructed by randomly perturbing these peaks using shuffleBed of BEDTools (version 2.28.0) (*Quinlan and Hall, 2010*). Based on the enrichment level, differential m⁶A peaks were identified as those with $\log_2$ |fold change| > $\log_2$ |1.5|.

## Statistics

Sample sizes were designed with adequate power according to the literature and our previous studies. No sample outliers were excluded. Experiments were not randomized, and the investigators were not blinded to allocation during experiments or outcome assessment. The variance between the groups being statistically compared was similar. Data are presented as mean ± SEM. Statistical analysis was performed using GraphPad Prism 8. All the data with n ≥ 6 was tested for normality using the Shapiro-Wilk normality test. For normally distributed data, comparisons between two groups were performed using unpaired Student's *t* test, and comparisons among three or more groups were performed using one-way or two-way ANOVA followed by Bonferroni's multiple comparisons correction; For non-normally distributed data and the data with n < 6, Mann-Whitney U test or the Kruskal-Wallis test followed by Dunn's multiple comparison tests were performed as appropriate. For the immunofluorescence images, quantification was normalized as interest of district per area. Three to five images per aorta and region of interest of each image were used for analysis. In all experiments, p<0.05 was considered statistically significant.

## Acknowledgements

We are grateful to Dr. Ding Ai and Dr. Yun-Gui Yang for critical discussion and technical support. IPA was utilized in the State Key Laboratory of Experimental Hematology, National Clinical Research Center for Blood Diseases. Funding This work was supported by grants from the National Natural Science Foundation of China (91940304, 81900396, 81870207, 82000477, 81970392), the National

Key R&D Program of China (2018YFA0801200), the Beijing Nova Program (Z201100006820104), and the Youth Innovation Promotion Association of Chinese Academy of Sciences (2018133). Bochuan Li is also funded by the Postdoctoral Science Foundation of China (2019M661041), the 'Postdoctoral Innovative Talent Support Program (BX20190235) and Excellent Sino-foreign Youth Exchange Program of China Association for Science and Technology.

## Additional information

### Funding

| Funder | Grant reference number | Author |
| --- | --- | --- |
| National Natural Science Foundation of China | 81900396 | Bochuan Li |
| Postdoctoral Research Foundation of China | 2019M661041 | Bochuan Li |
| Postdoctoral Research Foundation of China | BX20190235 | Bochuan Li |
| China Association for Science and Technology | Excellent Sino-foreign Youth Exchange Program | Bochuan Li |
| National Natural Science Foundation of China | 91940304 | Ying Yang |
| Chinese Academy of Sciences | 2018133 | Ying Yang |
| National Key Research and Development Program of China | 2018YFA0801200 | Ying Yang |
| Beijing Nova Program | Z201100006820104 | Ying Yang |
| National Natural Science Foundation of China | 81870207 | Yikui Tian |
| National Natural Science Foundation of China | 82000477 | Mengqi Li |
| National Natural Science Foundation of China | 81970392 | Hongfeng Jiang |

The funders had no role in study design, data collection and interpretation, or the decision to submit the work for publication.

### Author contributions

Bochuan Li, Formal analysis, Investigation, Writing – original draft; Ting Zhang, Mengxia Liu, Formal analysis, Software; Zhen Cui, Yanhong Zhang, Yanan Liu, Mengqi Li, Investigation; Mingming Liu, Investigation, Visualization; Yongqiao Sun, Methodology; Yikui Tian, Resources; Ying Yang, Hongfeng Jiang, Writing – review and editing; Degang Liang, Conceptualization, Funding acquisition, Supervision

### Author ORCIDs

Ying Yang http://orcid.org/0000-0002-8104-5985
Degang Liang http://orcid.org/0000-0003-2618-6651

### Ethics

The investigation conformed to the Guide for the Care and Use of Laboratory Animals by the US National Institutes of Health (NIH 17 Publication No. 85-23, revised in 2011). All study protocols and the use of animals were approved by the Institutional Animal Care and Use Committee of Tianjin Medical University.

### Decision letter and Author response

Decision letter https://doi.org/10.7554/eLife.69906.sa1

Author response https://doi.org/10.7554/eLife.69906.sa2

## Additional files

### Supplementary files

Supplementary file 1. Identified m$^6$A peaks in ST and OS conditions.

Supplementary file 2. Identified candidate genes with decreased m$^6$A and increased expression levels in response to OS.

Supplementary file 3. Information of primers used in this study.

Transparent reporting form

### Data availability

RNA-seq and MeRIP-seq data generated in this study have been deposited to the Gene Expression Omnibus under accession number GSE299805.

The following dataset was generated:

| Author(s) | Year | Dataset title | Dataset URL | Database and Identifier |
|---|---|---|---|---|
| Li B, Ting T, Liu M, Cui Z, Zhang Y, Liu M, Liu Y, Sun Y, Li M, Tian Y, Yang Y, Jiang H, Liang D | 2025 | RNA N6-methyladenosine modulates endothelial atherogenic responses to disturbed flow in mice | https://www.ncbi.nlm.nih.gov/geo/query/acc.cgi?acc=GSE299805 | NCBI Gene Expression Omnibus, GSE299805 |

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
