## [Editor Report]

Methylation of adenine residues in mRNA has been shown to be a regulator of many factors in heath and disease. In these studies, the authors present data that this modification of the mRNA for EGFR (epidermal growth factor receptor) through down regulation of the methylating enzyme METTL3 by shear stress is a contributor to vascular pathology in a model with some features of accelerated atherosclerosis.

---

## [Decision Letter]

**Decision letter after peer review:**

Thank you for submitting your article "RNA N6-methyladenosine modulates endothelial atherogenic responses to disturbed flow" for consideration by *eLife*. Your article has been reviewed by 3 peer reviewers, one of whom is a member of our Board of Reviewing Editors, and the evaluation has been overseen by Mone Zaidi as the Senior Editor. The reviewers have opted to remain anonymous.

Essential revisions:

You will see in the reviewers' comments numerous questions and suggestions. After consultation among the reviewers, we have listed the essential items that need to be addressed in a revised manuscript:

1) The evidence that mettl3 is downregulated with shear stress is not convincing. The western blot in Figure1a shows very subtle differences in mettl3 and many would question how the change proposed here modulates physiologic changes in m6A levels. To be convincing the authors should measure m6A levels as noted in Reviewer 2, point #1. This can be done using a commercial kit or preferably mass spec which is widely available from many groups now. Reviewer 3 also was concerned about m6A levels, and noted that the lack of m6A changes in vivo needs to be resolved, since altered m6A levels are the proposed mechanism of action.

2) The wild animals have almost undetectable mettl3 levels so they are practically

"Knockouts". Then what is the basis for the huge differences in the plaque burden?

5) What is the justification that the authors have to explain that all of the effects observed in AKT, ERK and VCAM-1 under OS are corrected by EGFR inhibition? The data supporting this conclusion is lacking.

6) The discussion of the Chien et al. PNAS 2021 is very narrow, especially considering the major discrepancies between the studies. The authors only note that the difference in the in vitro studies may be from shorter vs. longer exposure to shear stress, but his cannot be the explanation for the discrepant results in vivo using similar models. Thus, a full comparison needs to be part of the Discussion.

7) Also, there are a number of technical issues, including the lack of controls, the choice of HUVECS vs. more relevant aortic ECs, as well as statistical issues the reviewers have raised that need to be addressed.

*Reviewer #1:*

There is a lot of interesting information in this manuscript. Most of the studies in vivo have looked at METTL3 deficiency, so I think that the protective role of sustaining its expression in the ligation model should be explored in addition. Also, given the many RNAs found to be modified by METTL3, I find it surprising that it mainly comes down to EGFR in ECs, and that a major player is TSP1. TSP1, for example, has been investigated in atherosclerosis for its effects in two other major cells types in plaques, namely SMC and macrophages. Attributing all of its effects to what it does to ECs may reflect the model they use, which some consider as an "advanced atherosclerosis" process (it certainly has disturbed flow), but given that they have apoE-/- mice, a more standard atherosclerosis study could have been performed. Another technical note is that HUVECs are controversial in their direct relevance to arterial ECs. A final note is that there is a competing paper earlier this year in which some of the important results are discrepant, so there is a need for reconciling the findings.

The authors have provided a lot of detailed and interesting data to support their contention that the methylation of the RNA encoding EGFR is a major regulator of a vascular pathology related to atherosclerosis. There are some issues and gaps, however, in the story, including:

1. A lot of the studies are done in HUVECs. Historically, this cell model has been used for EC studies, but it is well appreciated that the direct relevance to arterial ECs is not clear. Were studies in other cell models of ECs performed, with similar results for key findings replicated?

2. TSP-1 has been studied for its effects on macrophages and SMCs, with reports to support its effects on atherosclerosis working through these cell types. The authors imply that all of the effects in their system is through the EC EGFR pathway. Is this perhaps a reflection of the ligation model they use? It has many notable differences from a standard model of atherosclerosis, so are the effects that seem to be confined to ECs model specific? With having apoE-/- mice available, I am surprised a standard atherosclerosis study was not performed.

3. The overexpression of METTL3 in vitro appeared to be protective against OS. While the authors provide experimental data in vivo with siRNA to further knock it down, I would think that an in vivo protection study would be a valuable addition. Also, with regard to the lentivirus experiment, how was it excluded that effects on other cell types (besides ECs) were not contributory?

4. THP-1 adhesion studies were done in some places, but the early studies referring to EC activation were based on the effects on adhesion molecules (e.g., VCAM). Some functional studies should have been done to show activation had a consequence on monocyte adhesion.

5. What is the basis for the effects of METTL3 changes on EGFR phosphorylation and the other signaling molecules reported on?

6. Going back to the specificity issue, there were many changes in the "methylome"- it is surprising that the major impact is exclusive to EGFR.

7. How do the authors explain the differences with the findings in Chien et al. (PNAS 118:e2025070118, 2021)?

*Reviewer #2:*

A notable strength of the current study is leveraging unbiased m6A interrogation approaches to decipher mechanisms of shear stress induced endothelial programming which is of substantial interest. Although others reported on generation of endothelial-specific Mettl3 KO, this is not a trivial endeavor and adds to the significance. It is difficult to ignore however that the most important conclusions of the paper were reported in a highly similar recent manuscript (Chien et al. PNAS. 2021). Notably the study from Shu Chien's group performed a rigorous interrogation of endothelial responses in response to oscillatory flow using eCLIP to map m6A sites, identified the reader protein involved and used a similar in vivo atherogenesis model perturbing Mettl3 to validate their findings. Implicating the EGFR signaling pathways and use of rescue/epistatic studies here is a novel aspect but the overall conceptual advance may be somewhat incremental.

1. The authors claim that mettl3 levels are reduced in response to OS but they provide no evidence that the subtle changes in mettl3 in endothelial cells(see figure 1A) translates in to meaningful changes in m6A levels. Thus, a major limitation of the work is that there are no direct m6A measurements? The authors need to measure global m6A levels preferably by mass spectrometry under different conditions. Measurement of m6A levels should also be done in the mettl3 overexpression studies.

2. Related to point#1, the proposed regulation of mettl3 in endothelial is opposite to previous work which showed that mettl3 is not downregulated and in fact upregulated in response to OS and other pro-atherogenic conditions (PMID: 33579825 and 32755566). The authors claim that these findings may be due to differences in conditions but this not well explored. The authors need to better consolidate their findings with previous work and ideally show experimentally the basis for Mettl3 regulation.

3. There are important details missing for key studies. How many replicates were used for the m6A-seq studies? How are the peaks shown in Figure 3C normalized? Can the authors show additional m6A peaks in the supplement? Can the authors also show a more expanded list of enriched motifs under different conditions and not just the top one?

4. Similar to the question above can the authors provide more details on number of replicates for RNA-seq study?

5. For some panels there appears to be dramatic differences in Mettl3 levels in the in vivo model. For example, in IHC staining shows that Mettl3 is almost completely absent in the LCA model (Figure 2D, and Supp Figure 5A). How do authors explain the big differences in plaque burden in Figure 5 if control animals are practically Mettl3 deficient? Curiously staining for Mettl3 is absent in figure 5.

6. The differences in motifs show in Figure 3B is an interesting finding but the physiologic significance of this finding is unclear since it invokes that the "interactome" of methyltransferase complex may be different depending on conditions which is not proven. Suggest minimizing this point in the text since its seems distracting.

7. Ref cited twice p28 line 7 and 10.

8. Please provide legends for abbreviations in figures.

*Reviewer #3:*

In the present work, Bochuan Li et al. studied the role of endothelial RNA N6-methyladenosine in atherogenesis. Using in vitro and in vivo approaches, Li and collaborators have shown that disturbed flow decreases the expression of methyltransferase METTL3 in endothelial cells. METTL3 is the enzyme responsible for nearly all the N6-methyladenosin (m6A) addition in mRNAs. Using in vitro methods, the authors show that under conditions of shear stress, human umbilical vein endothelial cells (HUVECs) demonstrate reduced m6A modification in the EGFR mRNA, which they associate with reduced EGFR mRNA degradation. Additionally, the authors used partial ligation of the left common carotid artery, which is an animal model of disturbed flow, to show that METTL3 expression is decreased with increased EGFR levels in the endothelium. They also demonstrate that the increase in EGFR expression promotes vascular inflammation and atherosclerosis by upregulating vascular adhesion molecule 1 (VCAM-1). These findings were also observed using a METTL3 endothelial cell specific knockout animal model. Furthermore, both overexpression of METTL3, or pharmacological inhibition of EGFR signaling, reduced VCAM-1 expression. Based on these results, the authors conclude that vascular inflammation in areas with disturbed flow is regulated by mRNA m6A modification, and that METTL3-mediated EGFR mRNA modification participates in the pathogenic mechanism of atherogenesis.

The major strength of this study is that the authors used a variety of in vitro and in vivo models, combining genetic approaches (METTL3 specific KO and the overexpression of METTL3), and pharmacological approaches, to show that there is a role for METTL3 and EGFR in atherogenesis.

Despite many of the strengths that this work has, there are several key weaknesses that make their conclusions less convincing and will require additional experimentation to resolve. In particular, there are significant differences between the effect of perturbed flow on METLL3 expression observed in vitro and in vivo. While in vivo data showed a dramatic reduction of METTL3, the in vitro data showed a modest reduction in enzyme levels, and there was no difference in m6A modification between cells treated with normal or disturbed flow. The lack of m6A changes is most problematic and needs to be resolved, since altered m6A levels are the proposed mechanism of action. Fundamentally, it is not clear whether the in vitro perturbed flow model recapitulates the key properties of in vivo perturbed flow, or whether the in vivo effect is due to features other than altered flow.

Another factor that makes it difficult to interpret these data concerns the way that overexpression or silencing by lentiviral vector infection was carried out, including a lack of control studies. There are no data demonstrating that overexpression or silencing only occurs in the endothelium layer (which itself is doubtful). Therefore, it is difficult to interpret these results when it is possible that there are contributions from other cell compartments within the aortic wall, including vascular smooth muscle cells and recruited leukocytes. To more fully understand the results of this study, a more detailed description of the methods is needed, and a greater discussion of the study's findings is necessary. In addition, it is not clear how the quantification of fluorescence was done. Importantly, a proper statistical analysis is critical to evaluate these findings and determine whether certain conclusions are warranted. Another issue that needs to be corrected concerns the small number of animals used in each experiment. With the small number of animals used, it is not certain if the data follows a normal distribution prior to applying a parametric analysis (T-test or ANOVA). It is also not clear whether it is appropriate to to use the standard error of the mean (SEM) rather than standard deviation (SD) in many of these studies. SD seems to be proper analysis based on descriptions in experimental methods.

The work presented by Li et al. is meaningful and has a potential to be published, but at the present time it is incomplete and will require additional experimentation and clarification.

1. It will be more relevant for the study if the in vitro experiments are performed on aortic endothelial cells rather than HUVECs, especially since the authors are comparing and making correlations between in vivo and in vitro studies. in vivo studies are of course investigating aortic endothelial cells.

2. The lack of effect on m6A modification between cells treated with normal or disturbed flow is disconcerting, since that it is the activity of METTL3 that is proposed to be the mediator of the effects in this study. Consequently, assuming these data are correct, the authors should examine if there is an increase in METTL16 that could account for the fact that with lower METTL3 levels there are no changes in m6A levels. Another possible explanation could be that the remaining METTL3 enzyme is sufficient to carry out methyltransferase activity, or that loss of m6A requires considerably greater amount of time than considered in these studies. In that case, it would be important to add a METTL3 knockdown study as a control. A METTL3 knockdown would help to determine which changes in mRNA expression are due to METTL3 downregulation in OS conditions.

3. In Figure 3F, it is important to show the effect of METTL3 knockdown on EGFR mRNA stability. It is possible that the increase in the EGFR mRNA under OS could be a transcriptional effect. It needs to be determined whether under OS conditions, EGFR mRNA has a slower decay rate.

4. The authors showed that OS increases EGFR expression and signaling in HUVECs. Additionally, they showed that the activation of EGFR under OS increases AKT and ERK phosphorylation and an increase in VCAM-1 expression. Overexpression of METTL3 avoids the effects previously described and silencing of METLL3 recapitulates OS effects, and inhibition of EGFR phosphorylation with AG1478 impaired AKT and ERK phosphorylation and VCAM-1 expression under OS conditions. However, it is not clear why VCAM-1 protein levels increase under OS or with METTL3 silencing, and this needs to be resolved. Is it NF-κB or AP-1 mediated? What is the justification that the authors have to explain that all of the effects observed in AKT, ERK and VCAM-1 under OS are corrected by EGFR inhibition? The data supporting this conclusion is lacking. The authors seem to be suggesting that OS is sensed by HUVECs in an EGFR-mediated manner exclusively? That seems difficult to believe, and needs to be supported by additional evidence. Also, METTL3 overexpression and silencing must be demonstrated by immunoblots in Figure 4.

5. For the experiments with partial ligation of the left common carotid artery in which METTL3 endothelial cell specific knockout animals were used, the authors need to provide representative images of the immunofluorescence in the RCA of KO animals. This should also be quantified and compared statistically.

6. There is considerable background in a number of the images that makes it difficult to distinguish signal from noise, especially for CD31 (Figure 5). Because of this issue, it is difficult to conclude that VCAM-1 colocalizes with CD31 positive cells. Proper unambiguous staining is critical to demonstrate that the increase in VCAM-1 is in the endothelium and it is not also increasing in other cells like vascular smooth muscle cells that under inflammatory conditions express VCAM-1 as well. The best approach is to carry out staining in cross sections like the HandE staining studies. That way it will be clear if VCAM-1 expression is occurring exclusively in the endothelium.

7. For all studies in which overexpression or silencing was carried out, the staining should be shown in cross sections to demonstrate that the silencing or overexpression is limited to the endothelium and not to the intima or the media of the aortic wall. Was the silencing or overexpression homogenous along the whole endothelium?

8. Perturbed flow occurs naturally along the aorta, as the authors mentioned. One of these areas is the aortic root, in which atherosclerotic plaques are formed in pro-atherogenic mouse models (like the animal model the authors used for this study). Thus, it will be very informative to show what happens in Apoe-/-EC-Mettl3KO versus Apoe-/-Mettl3flox/flox, with EGFR, METTL3 and VCAM-1 expression. What is the state of macrophage accumulation (CD68 staining), lipid accumulation (ORO staining), and vascular smooth muscle cell accumulation, and fibrous cap formation (ACTA2 staining) in the aortic root? This experiment will address the variability that is an issue in the partial ligation intervention studies, including local inflammation due to the procedure.

9. To use parametric tests like T-test or ANOVA, the data must follow a normal distribution. If data does not follow a normal distribution, then non-parametric tests should be carried out. Based on the methods section, it does not seem that the authors used any test to first check for data distribution, and therefore they may have applied the wrong statistical analysis. This point needs to be addressed. Also, the authors should justify why they represented their data as SEM instead as SD, especially for the animal experiments. They should explain in the methods section how they acquired the images and carried out quantification, with justification of the statistical analyses used. For the immunofluorescence studies, it will be necessary to normalize the intensity by area or by number of cells. Also, it will be important to describe how many images per aorta were used for analysis and the size of the areas analyzed by microscopy.

---

## [Author Response]

Essential revisions:You will see in the reviewers' comments numerous questions and suggestions. After consultation among the reviewers, we have listed the essential items that need to be addressed in a revised manuscript:1) The evidence that mettl3 is downregulated with shear stress is not convincing. The western blot in Figure1a shows very subtle differences in mettl3 and many would question how the change proposed here modulates physiologic changes in m6A levels. To be convincing the authors should measure m6A levels as noted in Reviewer 2, point #1. This can be done using a commercial kit or preferably mass spec which is widely available from many groups now. Reviewer 3 also was concerned about m6A levels, and noted that the lack of m6A changes in vivo needs to be resolved, since altered m6A levels are the proposed mechanism of action.

We appreciate this reviewer’s positive comments and valuable suggestions in general. In this revision, we have detected the level changes of m^6^A in HUVECs under ST and OS conditions using UHPLC-MRM-MS/MS (ultra-high-performance liquid chromatography-triple quadrupole mass spectrometry coupled with multiple-reaction monitoring), and observed significantly decreased m^6^A level in response to OS treatment. Consistent with the decreased METTL3 expression upon OS in HUVECs (Original Figure 1A-B; Revised Figure 1B-C), the expression of METTL3 was also decreased in response to OS in mouse aortic ECs (mAECs) (Revised Figure 1D-E). These data illustrated that METTL3-mediated m^6^A functions in the OS-induced endothelial atherogenic responses.

2) The wild animals have almost undetectable mettl3 levels so they are practically"Knockouts". Then what is the basis for the huge differences in the plaque burden?

Sorry for the misleading description. The METTL3 protein is decreased by 50% in LCA after partial ligation, undetectable METTL3 is only shown in EC-METTL3^KO^ mice (Original Figure 2D-E, Supplemental Figure 5).

5) What is the justification that the authors have to explain that all of the effects observed in AKT, ERK and VCAM-1 under OS are corrected by EGFR inhibition? The data supporting this conclusion is lacking.

Thanks for the valuable suggestion. Although EGFR is a key upstream factor of AKT and ERK signaling pathway, when treating samples under OS with EGFR inhibitor (AG-1478) to clearly elucidate its role in inflammatory response, we found that the phosphorylation of AKT, ERK, and VCAM-1 protein levels were decreased by AG-1478 under OS but still higher than the control level, suggesting that there are additional factors involving in the signaling pathway (Revised Figure 4E-H).

6) The discussion of the Chien et al. PNAS 2021 is very narrow, especially considering the major discrepancies between the studies. The authors only note that the difference in the in vitro studies may be from shorter vs. longer exposure to shear stress, but his cannot be the explanation for the discrepant results in vivo using similar models. Thus, a full comparison needs to be part of the Discussion.

To response this concern, we performed a standard atherosclerosis study in *Apoe^-/-^* EC-*Mettl3^KO^* mice and *Apoe^-/-^ Mettl3^flox/flox^* mice. ECs deficiency of METTL3 accelerated atherosclerotic lesion in aorta. Furthermore, ECs specific overexpression of METTL3 by AAV9 significantly reduced ECs activation. The overexpression or knockout experiments of METTL3 both in vivo and in vitro confirmed the pivotal role of METTL3 in endothelial atherogenic responses to disturbed flow (Revised Figures 6, 2F-G, and 5A-D).

7) Also, there are a number of technical issues, including the lack of controls, the choice of HUVECS vs. more relevant aortic ECs, as well as statistical issues the reviewers have raised that need to be addressed.

Thanks for the valuable suggestions. We have included complete control groups, added the mAECs experiment, and re-analyzed the statistics in the revised manuscripts and figures.

Reviewer #1:There is a lot of interesting information in this manuscript. Most of the studies in vivo have looked at METTL3 deficiency, so I think that the protective role of sustaining its expression in the ligation model should be explored in addition. Also, given the many RNAs found to be modified by METTL3, I find it surprising that it mainly comes down to EGFR in ECs, and that a major player is TSP1. TSP1, for example, has been investigated in atherosclerosis for its effects in two other major cells types in plaques, namely SMC and macrophages. Attributing all of its effects to what it does to ECs may reflect the model they use, which some consider as an "advanced atherosclerosis" process (it certainly has disturbed flow), but given that they have apoE-/- mice, a more standard atherosclerosis study could have been performed. Another technical note is that HUVECs are controversial in their direct relevance to arterial ECs. A final note is that there is a competing paper earlier this year in which some of the important results are discrepant, so there is a need for reconciling the findings.

We thank the reviewer for pointing out the importance of our work. We have followed the reviewer’s advice and performed additional experiments and analyses to provide further evidence for supporting our findings and conclusion.

First, we applied the AAV9-METTL3 OE in the ligation model in both EC-*Mettl3^KO^* mice and their littermates, and found specific overexpression of METTL3 in endothelial cell reversed the downregulated METTL3 level and upregulated EGFR and VCAM-1 levels in partial ligation model. (Revised Figures 2F-G, 5A-D, and S2C-D).

Second, we have explored the effect of TSP-1/EGFR in SMCs and macrophages in plaques, and found that knockdown of THBS1 by shRNA indeed reduced TSP-1 protein level in ECs, SMCs, and macrophages. However, the increased lesion induced by EC-METTL3^KO^ in *Apoe^-/-^* mice was returned to almost basal level after knockdown of THBS1 and EGFR, suggesting that TSP-1/EGFR mainly play a role in ECs in the partial ligation model.

Third, we have also carried out a standard atherosclerosis study in *Apoe^-/-^* EC-*Mettl3^KO^* mice and *Apoe^-/-^ Mettl3^flox/flox^* mice, and found ECs deficiency of METTL3 accelerated atherosclerotic lesion in aorta. (Revised Figure 6A-G).

These data have been included in the following point-to-point responses.

The authors have provided a lot of detailed and interesting data to support their contention that the methylation of the RNA encoding EGFR is a major regulator of a vascular pathology related to atherosclerosis. There are some issues and gaps, however, in the story, including:1. A lot of the studies are done in HUVECs. Historically, this cell model has been used for EC studies, but it is well appreciated that the direct relevance to arterial ECs is not clear. Were studies in other cell models of ECs performed, with similar results for key findings replicated?

Thanks for this very thoughtful comment. We have followed this suggestion to detect the levels of m^6^A and methyltransferase complex protein in HUVECs and mouse aortic ECs (mAECs). The results showed that similar as HUVECs, oscillatory shear stress (OS) induced a decreased METTL3 protein level in mAECs (Revised Figure 1D-E).

2. TSP-1 has been studied for its effects on macrophages and SMCs, with reports to support its effects on atherosclerosis working through these cell types. The authors imply that all of the effects in their system is through the EC EGFR pathway. Is this perhaps a reflection of the ligation model they use? It has many notable differences from a standard model of atherosclerosis, so are the effects that seem to be confined to ECs model specific? With having apoE-/- mice available, I am surprised a standard atherosclerosis study was not performed.

We thank this reviewer for her/his valuable comments. We emphasize TSP-1 is almost undetectable in ECs, but explosive increased under OS. Thus, we further measured TSP-1 protein level in plaques to explore the role of TSP-1 in ECs under partial ligation. The high expression of TSP-1 in LCA induced by partial ligation indeed decreased in ECs, SMCs, or macrophages after *shTHBS^-1^* treatment. It’s not surprising because TSP-1 is a secretory protein. However, *shTHBS1* returned the increased lesion induced by EC-METTL3^KO^ in *Apoe^-/-^* mice to almost basal level, suggesting that TSP-1/EGFR mainly affect in ECs in the partial ligation model (Author response image 1). Moreover, we increased the number of trials to clarify the role of EGFR (Revised Figure 4E-H), and found that inhibition of EGFR by AG1478 largely returns AKT and ERK phosphorylation to nearly basal level but still higher than control group. To confirm the results of partial ligation model, we led a standard atherosclerosis in *Apoe*^-/-^ EC-*Mettl3^KO^* mice and their littermates with 12 weeks’ western diet (Revised Figure 6A-G). The results showed that deficiency of METTL3 in ECs aggravate atherogenesis compared to control group in *Apoe^-/-^* mice.

**Author response image 1. sa2fig1:** TSP1 participates in endothelial atherogenic responses to disturbed flow. (A-B) 8-week-old male *Apoe^–/–^ Mettl3^flox/flox^* and *Apoe^–/–^* EC-*Mettl3^KO^* mice with 2 weeks of partial ligation were infused with the indicated lentiviruses, and arterial tissue cross sections immunofluorescence staining of the expression of TSP-1 in ECs (vWF) and macrophages (CD68) of the carotid artery of mice. Scale bar, 80 μm.

3. The overexpression of METTL3 in vitro appeared to be protective against OS. While the authors provide experimental data in vivo with siRNA to further knock it down, I would think that an in vivo protection study would be a valuable addition. Also, with regard to the lentivirus experiment, how was it excluded that effects on other cell types (besides ECs) were not contributory?

Thanks for the reviewer’s valuable suggestion. We have indeed proved that overexpressed adenovirus METTL3 has a protective role against ECs activation both in vitro and in vivo (Revised Figures 2F-G and S2C-D). Moreover, to exclude the influence of other cell types, we carried out AAV^endo^-METTL3 experiment in the partial ligation model, and found that specific overexpression of METTL3 in ECs provided a great protection in ECs activation (Revised Figures 2F-G and Supplemental Figure 2C-D). Through lentivirus experiment, we measured TSP-1 expression in ECs, SMCs and macrophages and observed significantly reduced TSP-1 upon lenti-shTHBS1 in ECs, SMCs, and macrophages in partial ligation model. Since TSP-1 is a secretory protein, it’s not remarkable that TSP-1 decreased in ECs, SMCs, and macrophages after *THBS1* knockdown. *THBS1* knockdown returned the increased lesion induced by EC-METTL3^KO^ in *Apoe^-/-^* mice to almost basal level, suggesting that TSP-1/EGFR mainly affect ECs in the partial ligation model (Author response image 1)

4. THP-1 adhesion studies were done in some places, but the early studies referring to EC activation were based on the effects on adhesion molecules (e.g., VCAM). Some functional studies should have been done to show activation had a consequence on monocyte adhesion.

Thanks for this suggestion. We have performed THP-1 adhesion studies under OS *vs.* ST in Original Supplemental Figure 4C-D (upper panel).

5. What is the basis for the effects of METTL3 changes on EGFR phosphorylation and the other signaling molecules reported on?

Thanks for the valuable suggestion. According to our existing data, the effects of METTL3 on EGFR and its downstream signaling depend on the post-transcriptional m^6^A modification. Under OS condition, METTL3 catalyzes the formation of m^6^A on *EGFR* mRNAs, which induced the subsequent *EGFR* mRNA degradation. The protein and phosphorylation levels of EGRR are the indirect consequence responding to METTL3 changes (as depicted in Revised Figure 4).

6. Going back to the specificity issue, there were many changes in the "methylome"- it is surprising that the major impact is exclusive to EGFR.

Thanks for the reviewer’s thoughtful comment. To respond to this concern, we have provided all the detected m^6^A-modified genes, with *EGFR* as one of relative top genes response to OS (Supplemental Table 2). Moreover, *EGFR* (Fold change of 0.872) was also identified as hypomethylated genes in Chien et al. (PNAS, 2021, Supplemental Table 1). We have added this discussion in the revised manuscript (line 10-11, page 15).

7. How do the authors explain the differences with the findings in Chien et al. (PNAS 118:e2025070118, 2021)?

In this article, we performed endothelial specific Mettl3 deficient mice with or without *Apoe^-/-^* background. Moreover, we carried out AAV9-METTL3 OE, not lentivirus, which cannot exclude the influence of other cells (such as SMCs and macrophages in aorta), to rescue the effect of EC activation in partial ligation model. The application of specific AAVs and knockdown mice allowed us to detect different phenomena in vivo and in vitro. Moreover, we and Chien all found that EGFR is a key molecular regulated by m^6^A.

Reviewer #2:A notable strength of the current study is leveraging unbiased m6A interrogation approaches to decipher mechanisms of shear stress induced endothelial programming which is of substantial interest. Although others reported on generation of endothelial-specific Mettl3 KO, this is not a trivial endeavor and adds to the significance. It is difficult to ignore however that the most important conclusions of the paper were reported in a highly similar recent manuscript (Chien et al. PNAS. 2021). Notably the study from Shu Chien's group performed a rigorous interrogation of endothelial responses in response to oscillatory flow using eCLIP to map m6A sites, identified the reader protein involved and used a similar in vivo atherogenesis model perturbing Mettl3 to validate their findings. Implicating the EGFR signaling pathways and use of rescue/epistatic studies here is a novel aspect but the overall conceptual advance may be somewhat incremental.

We thank the reviewer for the interests in our work. Just as the reviewer’s statement, we came across a very similar study with Chien et al. Otherwise, we identified a novel m^6^A-dependent METTL3-TSP-1-EGFR axis in regulating the atherogenic progression. In this story, we performed a standard atherosclerosis model in *ApoE^-/-^*EC*-Mettl3^KO^* mice (Revised Figure 6). As the effect of METTL3 in SMCs and macrophages in atherogenesis is largely unknown, we carried out endothelial specific overexpression of METTL3 using AAV9-METTL3 OE (adeno-associated virus) to focus on the function of METTL3 in ECs in vivo (Revised Figures 2F-G, 5A-D). Our work reveals an important role of RNA modification in atherosclerosis regulation.

1. The authors claim that mettl3 levels are reduced in response to OS but they provide no evidence that the subtle changes in mettl3 in endothelial cells(see figure 1A) translates in to meaningful changes in m6A levels. Thus, a major limitation of the work is that there are no direct m6A measurements? The authors need to measure global m6A levels preferably by mass spectrometry under different conditions. Measurement of m6A levels should also be done in the mettl3 overexpression studies.

We thank this reviewer for her/his valuable comments. We have followed the advice and measured Mettl3 expression and m^6^A levels in mouse aortic ECs (mAECs) and HUVECs in response to OS. As shown in Author response image 2; Revised Figure 1A, the m^6^A level was reduced in response to OS in HUVECs but not changed in mAECs. METTL3 protein in mAECs was significantly decreased in OS treatment, suggested that protein level of METTL3 is regulated by disturbed flow (Author response image 2, Revised Figure 1B-E).

**Author response image 2. sa2fig2:** . (A-C) UHPLC-MRM-MS analysis of m6A levels in mRNA extracted from HUVECs (A) and mAECs (C) exposed to ST and OS, and infected with the indicated adenoviruses (B). Data are shown as the mean ± SEM, *p<0.05, ns, not significant (Student’s *t* test). n=3.

2. Related to point#1, the proposed regulation of mettl3 in endothelial is opposite to previous work which showed that mettl3 is not downregulated and in fact upregulated in response to OS and other pro-atherogenic conditions (PMID: 33579825 and 32755566). The authors claim that these findings may be due to differences in conditions but this not well explored. The authors need to better consolidate their findings with previous work and ideally show experimentally the basis for Mettl3 regulation.

Thanks for the reviewer’s comment. We measured METTL3 expression and m^6^A level in HUVECs and mAECs (Author response image 1; Revised Figure 1). Similar with our previous results, the METTL3 and m^6^A levels are decreased in our experimental system.

3. There are important details missing for key studies. How many replicates were used for the m6A-seq studies? How are the peaks shown in Figure 3C normalized? Can the authors show additional m6A peaks in the supplement? Can the authors also show a more expanded list of enriched motifs under different conditions and not just the top one?

Thanks for the reviewer’s comment. We apologize for the missing description. Two biological replicates were used in the m^6^A-seq studies. The gene regions were split into windows and the peaks shown in Figure 3C were generated based on the ratio of reads mapped to IP and Input. The identified m^6^A peaks in ST and OS, and an expanded list of enriched motifs were shown in Supplemental Table 3.

4. Similar to the question above can the authors provide more details on number of replicates for RNA-seq study?

Thanks for this comment. Three replicates were used in RNA-seq studies. We have included the description and more detailed information in the revised manuscript (lines 9-10, page 20).

5. For some panels there appears to be dramatic differences in Mettl3 levels in the in vivo model. For example, in IHC staining shows that Mettl3 is almost completely absent in the LCA model (Figure 2D, and Supp Figure 5A). How do authors explain the big differences in plaque burden in Figure 5 if control animals are practically Mettl3 deficient? Curiously staining for Mettl3 is absent in figure 5.

We are sorry for the misleading description. In original Figure 2D and Supplemental Figure 5A, Mettl3 levels in LCA decreased about 50% compared with RCA in Mettl3^flox/flox^ mice. While Mettl3 almost completely absent in EC-*Mettl3^KO^* mice just shows its knockout efficiency. The staining of Mettl3 has now been shown in Supplemental Figure 5A

6. The differences in motifs show in Figure 3B is an interesting finding but the physiologic significance of this finding is unclear since it invokes that the "interactome" of methyltransferase complex may be different depending on conditions which is not proven. Suggest minimizing this point in the text since its seems distracting.

Thanks for this suggestion. We agree with the points that there are many factors including “writers” and “erasers” involving in the dynamic m^6^A regulation and the interactome between the factors are different under different conditions. We have followed the advice to minimize the motif difference in the revised manuscript (lines 9-10, page 7).

7. Ref cited twice p28 line 7 and 10.

We appreciate the suggestion and have corrected it in the revised manuscript.

8. Please provide legends for abbreviations in figures.

Thanks for the suggestion and we have provided legends abbreviations in the revised figures.

Reviewer #3:In the present work, Bochuan Li et al. studied the role of endothelial RNA N6-methyladenosine in atherogenesis. Using in vitro and in vivo approaches, Li and collaborators have shown that disturbed flow decreases the expression of methyltransferase METTL3 in endothelial cells. METTL3 is the enzyme responsible for nearly all the N6-methyladenosin (m6A) addition in mRNAs. Using in vitro methods, the authors show that under conditions of shear stress, human umbilical vein endothelial cells (HUVECs) demonstrate reduced m6A modification in the EGFR mRNA, which they associate with reduced EGFR mRNA degradation. Additionally, the authors used partial ligation of the left common carotid artery, which is an animal model of disturbed flow, to show that METTL3 expression is decreased with increased EGFR levels in the endothelium. They also demonstrate that the increase in EGFR expression promotes vascular inflammation and atherosclerosis by upregulating vascular adhesion molecule 1 (VCAM-1). These findings were also observed using a METTL3 endothelial cell specific knockout animal model. Furthermore, both overexpression of METTL3, or pharmacological inhibition of EGFR signaling, reduced VCAM-1 expression. Based on these results, the authors conclude that vascular inflammation in areas with disturbed flow is regulated by mRNA m6A modification, and that METTL3-mediated EGFR mRNA modification participates in the pathogenic mechanism of atherogenesis.The major strength of this study is that the authors used a variety of in vitro and in vivo models, combining genetic approaches (METTL3 specific KO and the overexpression of METTL3), and pharmacological approaches, to show that there is a role for METTL3 and EGFR in atherogenesis.Despite many of the strengths that this work has, there are several key weaknesses that make their conclusions less convincing and will require additional experimentation to resolve. In particular, there are significant differences between the effect of perturbed flow on METLL3 expression observed in vitro and in vivo. While in vivo data showed a dramatic reduction of METTL3, the in vitro data showed a modest reduction in enzyme levels, and there was no difference in m6A modification between cells treated with normal or disturbed flow. The lack of m6A changes is most problematic and needs to be resolved, since altered m6A levels are the proposed mechanism of action. Fundamentally, it is not clear whether the in vitro perturbed flow model recapitulates the key properties of in vivo perturbed flow, or whether the in vivo effect is due to features other than altered flow.Another factor that makes it difficult to interpret these data concerns the way that overexpression or silencing by lentiviral vector infection was carried out, including a lack of control studies. There are no data demonstrating that overexpression or silencing only occurs in the endothelium layer (which itself is doubtful). Therefore, it is difficult to interpret these results when it is possible that there are contributions from other cell compartments within the aortic wall, including vascular smooth muscle cells and recruited leukocytes. To more fully understand the results of this study, a more detailed description of the methods is needed, and a greater discussion of the study's findings is necessary. In addition, it is not clear how the quantification of fluorescence was done. Importantly, a proper statistical analysis is critical to evaluate these findings and determine whether certain conclusions are warranted. Another issue that needs to be corrected concerns the small number of animals used in each experiment. With the small number of animals used, it is not certain if the data follows a normal distribution prior to applying a parametric analysis (T-test or ANOVA). It is also not clear whether it is appropriate to to use the standard error of the mean (SEM) rather than standard deviation (SD) in many of these studies. SD seems to be proper analysis based on descriptions in experimental methods.

We sincerely thank the reviewer for the careful review. In the revision, we measured m^6^A levels in HUVECs and mAECs under disturbed flow (Author revised image 1; Revised Figure 1). The in vitro perturbed flow model can simulate the properties of in vivo perturbed flow as we and Chien reported. in vivo experiments, we carried out endothelial specific METTL3 deficient mice, and AAV9-METTL3 OE to induce overexpressing METTL3 in ECs (Revised Figures 2F-G, 5A-D, 6). At last, we completed method information as the reviewer suggested.

The work presented by Li et al. is meaningful and has a potential to be published, but at the present time it is incomplete and will require additional experimentation and clarification.1. It will be more relevant for the study if the in vitro experiments are performed on aortic endothelial cells rather than HUVECs, especially since the authors are comparing and making correlations between in vivo and in vitro studies. in vivo studies are of course investigating aortic endothelial cells.

We thank this reviewer for her/his valuable comments. In the revision, we used mouse aortic ECs (mAECs) to confirm our discovery in HUVECs. Similar as the HUVECs results, mAECs measurement showed decreased Mettl3 levels in response to OS (Revised Figure 1B-E).

2. The lack of effect on m6A modification between cells treated with normal or disturbed flow is disconcerting, since that it is the activity of METTL3 that is proposed to be the mediator of the effects in this study. Consequently, assuming these data are correct, the authors should examine if there is an increase in METTL16 that could account for the fact that with lower METTL3 levels there are no changes in m6A levels. Another possible explanation could be that the remaining METTL3 enzyme is sufficient to carry out methyltransferase activity, or that loss of m6A requires considerably greater amount of time than considered in these studies. In that case, it would be important to add a METTL3 knockdown study as a control. A METTL3 knockdown would help to determine which changes in mRNA expression are due to METTL3 downregulation in OS conditions.

Thanks for the reviewer’s valuable suggestions. We have followed the advice and measured m^6^A levels by UHPLC-MRM-MS/MS in HUVECs and mAECs (Author response image 1; Revised Figure 1A). Furthermore, we detected METTL16 protein levels in control and METTL3 deficient cells and found that METTL16 remains unchanged upon Mettl3 knockdown (Author response image 3).

**Author response image 3. sa2fig3:** METTL16 remains unchanged upon Mettl3 knockdown. HUVECs and mAECs were infected with METTL3 siRNA for 24 hr, Western blot analysis of METTL3, METTL16 and GAPDH. (B and D) Quantification of the expression of the indicated proteins in (A and C). Data are shown as the mean ± SEM, *p<0.05, NS, not significant (Student’s *t* test). n=6.

3. In Figure 3F, it is important to show the effect of METTL3 knockdown on EGFR mRNA stability. It is possible that the increase in the EGFR mRNA under OS could be a transcriptional effect. It needs to be determined whether under OS conditions, EGFR mRNA has a slower decay rate.

We thank this reviewer for her/his valuable comments. We have followed this advice and detected mRNA stability of *EGFR* under METTL3 knockdown or OS treatment. As expected, *EGFR* mRNA showed a slower decay rate response to OS or METTL3 knockdown (Revised Figure 3G-H).

4. The authors showed that OS increases EGFR expression and signaling in HUVECs. Additionally, they showed that the activation of EGFR under OS increases AKT and ERK phosphorylation and an increase in VCAM-1 expression. Overexpression of METTL3 avoids the effects previously described and silencing of METLL3 recapitulates OS effects, and inhibition of EGFR phosphorylation with AG1478 impaired AKT and ERK phosphorylation and VCAM-1 expression under OS conditions. However, it is not clear why VCAM-1 protein levels increase under OS or with METTL3 silencing, and this needs to be resolved. Is it NF-κB or AP-1 mediated? What is the justification that the authors have to explain that all of the effects observed in AKT, ERK and VCAM-1 under OS are corrected by EGFR inhibition? The data supporting this conclusion is lacking. The authors seem to be suggesting that OS is sensed by HUVECs in an EGFR-mediated manner exclusively? That seems difficult to believe, and needs to be supported by additional evidence. Also, METTL3 overexpression and silencing must be demonstrated by immunoblots in Figure 4.

We agree with this valuable comment. EGFR/ERK/AKT induced EC activation *via* NF-κB signaling, with VCAM-1 as adhesion molecular regulated by NF-κB as reported. To explore the role of EGFR in response to OS in HUVECs, we increased sample numbers and found that METTL3 overexpression or EGFR inhibition largely reversed the effects of OS on AKT, ERK and VCAM-1 expression (Revised Figure 4E-H).

5. For the experiments with partial ligation of the left common carotid artery in which METTL3 endothelial cell specific knockout animals were used, the authors need to provide representative images of the immunofluorescence in the RCA of KO animals. This should also be quantified and compared statistically.

We appreciate the suggestion and add the immunofluorescent images of RCA in knockout model.

6. There is considerable background in a number of the images that makes it difficult to distinguish signal from noise, especially for CD31 (Figure 5). Because of this issue, it is difficult to conclude that VCAM-1 colocalizes with CD31 positive cells. Proper unambiguous staining is critical to demonstrate that the increase in VCAM-1 is in the endothelium and it is not also increasing in other cells like vascular smooth muscle cells that under inflammatory conditions express VCAM-1 as well. The best approach is to carry out staining in cross sections like the HandE staining studies. That way it will be clear if VCAM-1 expression is occurring exclusively in the endothelium.

Thanks for the suggestion. To respond this concern, we provided cross section of LCA from ligation model. The results showed an unambiguous staining of VCAM-1, EGFR, and TSP-1 colocalize with vWF (another endothelial cell marker) (Revised Figure 2F-G and 5A-D), and excluded function of other cell type like smooth muscle cells and macrophages by using AAV9-METTL3 OE in our model (Revised Supplemental Figure 2C-D).

7. For all studies in which overexpression or silencing was carried out, the staining should be shown in cross sections to demonstrate that the silencing or overexpression is limited to the endothelium and not to the intima or the media of the aortic wall. Was the silencing or overexpression homogenous along the whole endothelium?

We appreciate the suggestion and complete it in the revision. To avoid the function of SMCs and other cells, we carried out endothelial specific overexpression of METTL3 using AAV9-METTL3 OE (adeno-associated virus) to focus on the function of METTL3 in ECs in vivo. We have included these results in (Revised Figure 2F-G and 5A-D, Supplemental Figure 2C-D).

8. Perturbed flow occurs naturally along the aorta, as the authors mentioned. One of these areas is the aortic root, in which atherosclerotic plaques are formed in pro-atherogenic mouse models (like the animal model the authors used for this study). Thus, it will be very informative to show what happens in Apoe-/-EC-Mettl3KO versus Apoe-/-Mettl3flox/flox, with EGFR, METTL3 and VCAM-1 expression. What is the state of macrophage accumulation (CD68 staining), lipid accumulation (ORO staining), and vascular smooth muscle cell accumulation, and fibrous cap formation (ACTA2 staining) in the aortic root? This experiment will address the variability that is an issue in the partial ligation intervention studies, including local inflammation due to the procedure.

Thanks for the suggestion. We performed serious of lesion analysis and showed that *Apoe^-/-^*EC*-Mettl3^KO^* mice significant exacerbate the lesion area and compared with control mice (Revised Figure 6).

9. To use parametric tests like T-test or ANOVA, the data must follow a normal distribution. If data does not follow a normal distribution, then non-parametric tests should be carried out. Based on the methods section, it does not seem that the authors used any test to first check for data distribution, and therefore they may have applied the wrong statistical analysis. This point needs to be addressed. Also, the authors should justify why they represented their data as SEM instead as SD, especially for the animal experiments. They should explain in the methods section how they acquired the images and carried out quantification, with justification of the statistical analyses used. For the immunofluorescence studies, it will be necessary to normalize the intensity by area or by number of cells. Also, it will be important to describe how many images per aorta were used for analysis and the size of the areas analyzed by microscopy.

Thanks. We corrected the sentences as “All the data with N≥6 was tested for normality using the Shapiro-Wilk normality test. For normally distributed data, comparisons between 2 groups were performed using unpaired Student’s t-test, and comparisons among 3 or more groups were performed using one-way or two-way ANOVA followed by Bonferroni's multiple comparisons correction; For non-normally distributed data and the data with N<6, Mann-Whitney U test or the Kruskal-Wallis test followed by Dunn’s multiple comparison tests were performed as appropriate. For the immunofluorescence images, quantification was normalized as interest of district per area (IOD/area). Three-five images per aorta and ROI (region of interest) of each image were used for analysis.” on Page 22 line 12-20 in the revised manuscript.

SEM and SD are indicators reflecting the degree of variation to represent statistic data, the data were statistically significant when represent as mean ± SD.